# RAC1B modulates intestinal tumourigenesis via modulation of WNT and EGFR signalling pathways

Victoria Gudiño[1,10], Sebastian Öther-Gee Pohl [1], Caroline V. Billard[1], Patrizia Cammareri[1], Alfonso Bolado[1], Stuart Aitken [2,3], David Stevenson[4], Adam E. Hall[1], Mark Agostino [5,6], John Cassidy[7], Colin Nixon [4], Alex von Kriegsheim [1], Paz Freile[1,2,3], Linda Popplewell[8], George Dickson[8], Laura Murphy[3], Ann Wheeler [3], Malcolm Dunlop[1,2,3], Farhat Din[1,2,3], Douglas Strathdee [4], Owen J. Sansom [4,9] & Kevin B. Myant [1✉]

Current therapeutic options for treating colorectal cancer have little clinical efficacy and acquired resistance during treatment is common, even following patient stratification. Understanding the mechanisms that promote therapy resistance may lead to the development of novel therapeutic options that complement existing treatments and improve patient outcome. Here, we identify RAC1B as an important mediator of colorectal tumourigenesis and a potential target for enhancing the efficacy of EGFR inhibitor treatment. We find that high *RAC1B* expression in human colorectal cancer is associated with aggressive disease and poor prognosis and deletion of *Rac1b* in a mouse colorectal cancer model reduces tumourigenesis. We demonstrate that RAC1B interacts with, and is required for efficient activation of the EGFR signalling pathway. Moreover, RAC1B inhibition sensitises cetuximab resistant human tumour organoids to the effects of EGFR inhibition, outlining a potential therapeutic target for improving the clinical efficacy of EGFR inhibitors in colorectal cancer.

[1] Cancer Research UK Edinburgh Centre, MRC Institute of Genetics & Molecular Medicine, The University of Edinburgh, Western General Hospital, Edinburgh EH4 2XU, UK. [2] MRC Human Genetics Unit, MRC Institute of Genetics & Molecular Medicine, The University of Edinburgh, Western General Hospital, Edinburgh EH4 2XU, UK. [3] Institute of Genetics and Molecular Medicine, University of Edinburgh, Western General Hospital, Edinburgh EH4 2XU, UK. [4] Cancer Research UK Beatson Institute, Garscube Estate, Bearsden, Glasgow G61 1BD, UK. [5] School of Pharmacy and Biomedical Sciences, Curtin Health and Innovation Research Institute, Curtin University, Perth, WA 6845, Australia. [6] Curtin Institute for Computation, Curtin University, Perth, WA 6845, Australia. [7] Cancer Research UK Cambridge Institute, University of Cambridge, Li Ka Shing Centre, Cambridge CB2 0RE, UK. [8] School of Biological Sciences, Royal Holloway - University of London, Egham, Surrey TW20 0EX, UK. [9] Institute of Cancer Sciences, University of Glasgow, Garscube Estate, Bearsden, Glasgow G61 1QH, UK. [10] Present address: Inflammatory Bowel Disease Unit, Department of Gastroenterology, Institut d'Investigacions Biomèdiques August Pi i Sunyer (IDIBAPS) - CIBEREHD, Barcelona, Spain. ✉email: kevin.myant@igmm.ed.ac.uk

olorectal cancer (CRC) is the second commonest cause of cancer-related mortality with highly variable disease outcome and response to therapies. Part of these differences are believed to be due to extensive genomic and transcriptomic heterogeneity leading to the acquisition of rapid resistance to available treatments[1]. For example, the effectiveness of the anti-epidermal growth factor (EGF) antibodies cetuximab or panitumumab, which are utilised in a subgroup of *KRAS* wild-type (WT) CRCs is limited, with resistance rapidly emerging via multiple and diverse mechanisms[2,3]. Recent efforts to understand CRC complexity has led to the definition of a number of distinct consensus molecular subtypes (CMSs) based on gene expression patterns of tumour biopsies or purified tumour epithelial cells[1,4]. These different subtypes classify tumours with distinguishing features such as microsatellite instability (MSI), high/hypermutated/*BRAF* mutated (CMS1), WNT activated (CMS2), metabolic/*KRAS* mutated (CMS3) and EMT/TGF-β activated (CMS4). Classification into different subtypes provides useful predictive information such as patient prognosis and predicted response to therapies. For example, patients with CMS2 tumours preferentially benefit from anti-epidermal growth factor receptor (EGFR) and anti-vascular endothelial growth factor (VEGF) therapy[5,6]. However, even within CMS subtypes extensive heterogeneity exists and despite stratification many patients benefit only from an initial treatment response with therapy resistance frequently observed. Interestingly, recent evidence suggests that resistance to the EGFR inhibitor cetuximab can be acquired by switching molecular subtype indicating non-genetic, transcriptional mechanisms may play an important role in modulating response to therapy[7]. Together, this outlines a pressing need to develop both novel therapeutic options and to also better understand disease complexity to enable better stratification of available treatments.

The majority of CRC cases are initiated by loss or inactivation of the *APC* tumour suppressor gene with accumulated mutations in other key pathways, such as MAPK, TP53 and TGF-β promoting tumour progression[8–10]. APC is a negative regulator of the WNT signalling pathway that when lost, allows β-catenin to accumulate in the nucleus and drive an oncogenic transcription programme leading to tumour formation[11,12]. Previous studies have identified the WNT target gene *Myc* as a key mediator of oncogenic WNT signalling and demonstrated reduced MYC levels perturbs intestinal tumorigenesis[13,14]. MYC has long been proposed as a therapeutic target for multiple cancer types but direct inhibition of the protein has proven difficult owing to a lack of defined ligand binding sites. However, a number of pathways regulated by MYC signalling have subsequently been shown to be important for efficient tumorigenesis following *APC* loss demonstrating alternative mechanisms by which the outputs of oncogenic WNT signalling may be targeted[15–20].

We previously identified one such pathway, RAC1 signalling, as being critical for the expansion of intestinal stem cells and subsequent tumour formation following *Apc* deletion in the mouse[19]. Activation of RAC1 is achieved by binding of Rho-Guanine Exchange factors (GEFs) and we previously identified upregulation (and subsequent RAC1 activation) of a number of these following *Apc* loss[19]. An alternative mechanism via which RAC1 signalling can be activated is via the splice variant termed RAC1B, which is overexpressed in numerous tumour types[21]. RAC1B results from the inclusion of exon 4 (alternatively designated exon 3b) encoding an additional 19 amino acids which leads to constitutive activation[22–24]. It is believed that RAC1B has a distinctive, more restricted set of effector pathways than RAC1, but appears to be more critical for tissue transformation[25–29]. Despite some in vitro evidence supporting a tumorigenic role for RAC1B, its in vivo function and mechanism of action is poorly understood and to date

no studies have assessed whether RAC1B is required for tumorigenesis in vivo and thus the potential benefits of its therapeutic targeting.

We therefore set out to determine the requirement for RAC1B during intestinal tumorigenesis. Here we find that *RAC1B* is overexpressed in CRC and high *RAC1B* expression correlates with high WNT activity and poor prognosis. We find that deletion of *Rac1b* in a mouse model of intestinal cancer significantly increases survival and reduces tumour number, tumour-cell proliferation and tumorigenic WNT signalling. Mechanistically, RAC1B interacts with a network of membrane-bound receptor tyrosine kinases (RTKs) including the important oncogenic mediators EGFR and ERBB2. RAC1B is required for efficient activation of EGFR signalling and organoid cultures lacking *Rac1b* are sensitised to EGFR inhibition. Importantly, cetuximab-resistant human tumour organoids treated with a novel inhibitor of *RAC1B* splicing show this same increased sensitivity to EGFR inhibition suggesting RAC1B may be a candidate therapeutic target for co-treatment with EGFR inhibitors in CRC.

## Results

***RAC1B* overexpression correlates with high WNT activity and poor prognosis in human CRC**. To investigate the expression of *RAC1B* in human CRC, we analysed previously determined percent spliced in (PSI) values of *RAC1B* exon 4 from TCGA RNAseq experiments[30]. In line with previous reports, this demonstrated a significant increase in *RAC1B* expression in human colorectal tumours (Fig. 1A). We confirmed this using isoform expression data obtained directly from TCGA RNAseq dataset (Fig. S1A). We also observed significantly increased *RAC1B* expression in advanced stage (TIII and TIV) compared to early stage (TI and TII) disease, in patients with lymphovascular tumour invasion and in patients with metastatic disease (Fig. 1B). Tumours were classified as high or low for *RAC1* expression taking as reference the thresholds for PSI values described previously[21]. Using a stringent threshold we designated three groups: *RAC1B*^high (delta PSI > 0.2 above normal tissue average, ~18% of tumours), *RAC1B*^low (PSI value lower than normal tissue average, ~7% of tumours) and *RAC1B*^int (intermediate expression, ~75% of tumours) (Fig. S1B). Confirming our expression analysis, tumours in the *RAC1B*^high group were significantly more likely to be found at a more advanced stage and with presence of lymphovascular invasion than tumours in the *RAC1B*^low and *RAC1B*^int groups (Fig. S1C, D). Furthermore, high levels of *RAC1B* expression were correlated with significantly reduced disease-free (hazard ratio of 2.64) and lower overall survival (hazard ratio of 2.08) (Fig. 1C). These data demonstrate that elevated levels of *RAC1B* correlates with poorer disease outcome in CRC.

To determine whether high *RAC1B* expression was associated with a particular CRC CMS, we correlated expression of *RAC1B* with the gene classifier sets for each individual subtype. *RAC1B* expression was positively correlated with the majority of genes that classify the CMS2 subtype and negatively correlated with the majority of genes that classify the CMS1 subtype (Figs. 1D and S1E). Little enrichment for either the CMS3 or the CMS4 subtype was observed. Further analysis of the transcriptional profile of *RAC1B* in this dataset confirmed these observations with tumours classified as CMS1 expressing low levels of *RAC1B* and tumours classified as CMS2 expressing high levels of *RAC1B* (Figs. 1E and S1F). Additionally, we found that *RAC1B*^high tumours were most common in CMS2 tumours, although notably also present in CMS4 (Fig. S1G, H). Furthermore, GSEA carried out on *RAC1B*^high vs *RAC1B*^low expressing tumours confirmed the positive enrichment of genes associated with the CMS2 subtype,

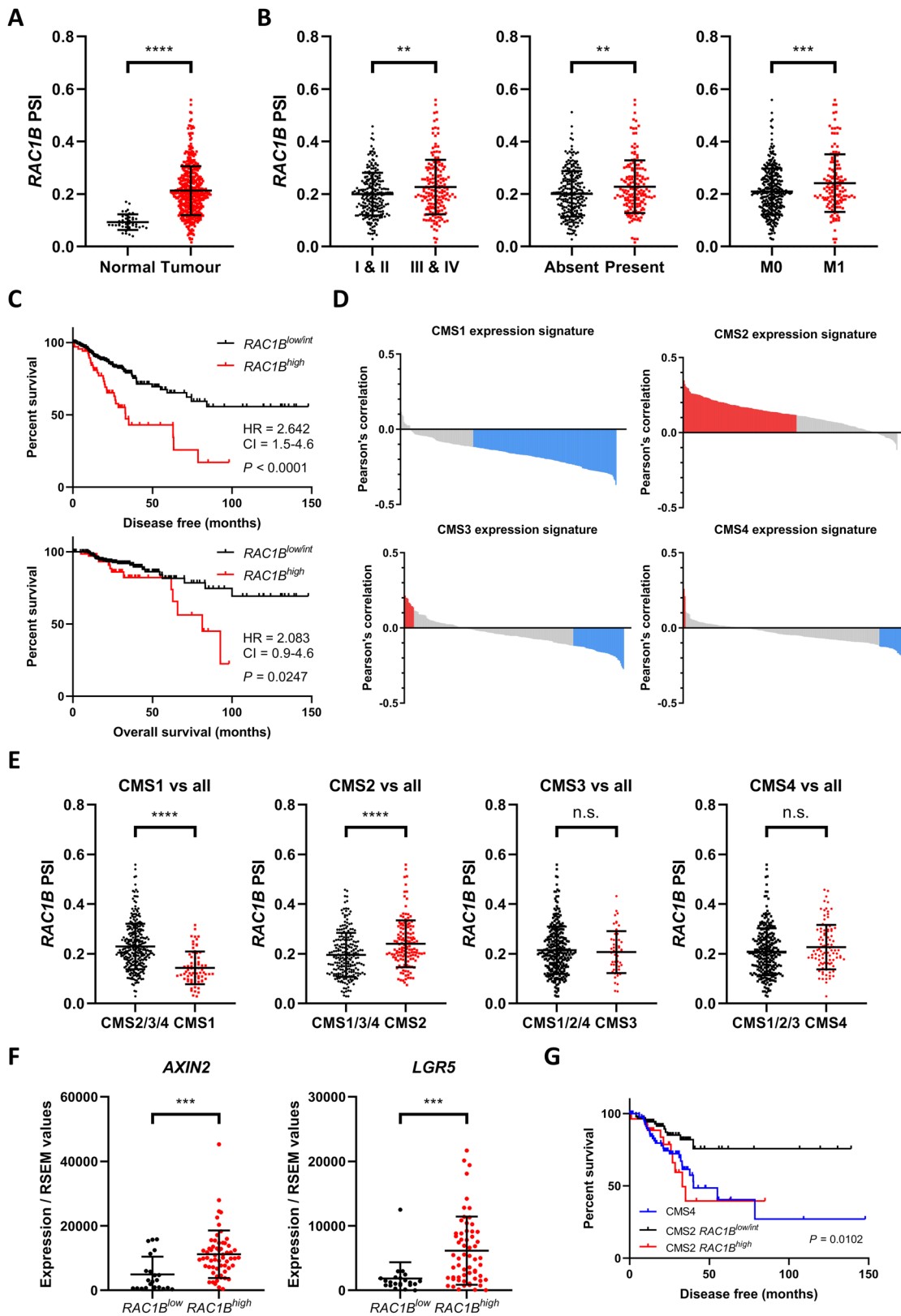

other gene sets associated with WNT signalling activation and a negative enrichment of genes associated with the CMS1 subtype (Fig. S1I). Validating these findings, we found expression of canonical WNT target genes *AXIN2, LGR5, TCF7* and *ASCL2* were elevated in tumours expressing high levels of *RAC1B* (Figs. 1F and S1J). The same analysis revealed significantly lower

expression of *RAC1B* in MSI tumours and/or tumours carrying mutations in *BRAF* as hypothesised by the negative correlation with CMS1 (Fig. S1K). These data indicate *RAC1B* expression correlates with high levels of WNT signalling in human CRC. Apparently contradictory to these findings, the prognosis of patients with WNT-activated CMS2 tumours is relatively good

**Fig. 1 *RAC1B* overexpression correlates with tumour progression and poor prognosis in human CRC. A** Individual value plot of *RAC1B* exon 4 PSI values for normal and colorectal tumour tissue (TCGA dataset) (data are presented as mean ± SD; $^{****}P = 2 \times 10^{-15}$; two-tailed *t*-test, $n = 41$ vs 457 tumour samples). **B** Individual value plots of *RAC1B* exon 4 PSI values from TCGA CRC dataset comparing: early (TI&II) vs late (TIII&TIV) stage CRC (left panel) (data are presented as mean ± SD; $^{**}P = 0.0029$; two-tailed *t*-test, $n = 227$ vs 179 tumour samples), absence vs presence of lymphovascular invasion (middle panel) (data are presented as mean ± SD; $^{**}P = 0.004$; two-tailed *t*-test, $n = 241$ vs 162 tumour samples) and absence (M0) or presence (M1) of metastasis (right panel) (data are presented as mean ± SD; $^{***}P = 0.0008$; two-tailed *t*-test, $n = 329$ vs 136 tumour samples). **C** Disease-free (top panel) and overall (bottom panel) survival of human CRC patients separated by high vs low/intermediate *RAC1B* expression. Log-rank *P* value, hazard ratio (HR) and 95% confidence intervals (CI) presented. **D** Bar plots of Pearson's correlation coefficients for *RAC1B* exon 4 PSI values and genes that identify different CRC CMSs. Each bar corresponds to the correlation of *RAC1B* and an individual gene, red bars indicate genes positively correlated with *RAC1B* and blue bars genes negatively correlated with *RAC1B* expression ($P < 0.05$). **E** Individual value plots of *RAC1B* exon 4 PSI values in tumours from each CMS compared to all other tumours. CMS1 vs all; data are presented as mean ± SD; $^{****}P = 4.83 \times 10^{-13}$; two-tailed *t*-test, $n = 71$ vs 299 tumour samples. CMS2 vs all; data are presented as mean ± SD; $^{****}P = 5.49 \times 10^{-6}$; two-tailed *t*-test, $n = 145$ vs 225 tumour samples. CMS3 vs all; data are presented as mean ± SD, $n = 54$ vs 316 tumour samples. CMS4 vs all; data are presented as mean ± SD, $n = 100$ vs 270 tumour samples. **F** Individual value plots of WNT target gene expression in tumours expressing low vs high *RAC1B*. *AXIN2*; data are presented as mean ± SD; $^{***}P = 0.0005$; two-tailed *t*-test, $n = 22$ vs 60 tumour samples. *LGR5*; data are presented as mean ± SD; $^{***}P = 0.0004$; two-tailed *t*-test, $n = 22$ vs 60 tumour samples. **G** Disease-free survival of human CRC patients with tumours from the CMS2 CRC subtype separated by high vs low/intermediate *RAC1B* expression. Survival of patients with CMS4 CRC subtype tumours included for comparison. Log-rank *P* value shown. Source data are provided as a Source Data file.

with CMS4 tumours resulting in the worst overall survival[1]. As *RAC1B*[high] tumours are also found in CMS4 it is possible that these tumours are responsible for the poor prognosis identified for *RAC1B*[high] tumours. To investigate this further, we plotted the outcome of patients with CMS2 tumours only, split according to *RAC1B*-expression levels. This analysis demonstrated that patients with CMS2 tumours could be separated based on *RAC1B* expression with high levels of *RAC1B* conferring survival outcomes as poor as patients with the highly aggressive mesenchymal CMS4 subtype tumours (Fig. 1G). Therefore, a significant proportion of CMS2 tumours have poor prognosis and these are marked by high levels of *RAC1B* expression. Together, these data demonstrate that high *RAC1B* expression marks a subset of tumours with high WNT activity and poor patient outcome suggesting an important role in colorectal tumorigenesis.

***Rac1b* is expressed in mouse intestinal crypts and its expression is increased following *Apc* deletion.** We next examined the expression of *Rac1b* in mouse intestinal tissue. To do so, we fractioned intestinal epithelium into villus and crypt / crypt base fractions (Fig. 2A). Expression analysis of stem and progenitor cell markers validated the fractionation protocol (Fig. S2A). Interestingly, whilst *Rac1* was fairly uniformly expressed across all fractions, *Rac1b* showed significantly higher expression in the crypt fractions, indicative of high expression in proliferative intestinal cells (Figs. 2B and S2A). To investigate whether *Rac1b* expression is enriched in intestinal stem cells, we sorted Lgr5+ cells from mice carrying the *Lgr5-EGFP-ires-CreERT2* transgene (Fig. S2B). Consistent with our fractionation experiments, we found increased *Rac1b* expression in Lgr5+ compared to Lgr5- cells (Fig. 1C). To confirm these observations we designed a RNA in situ hybridisation BaseScope probe to detect *Rac1b* expression (exon 4–5 junction) (Figs. 2D and S2C). We detected low *Rac1b*-expression levels in mouse small intestine (SI) and large intestine (LI) with significantly higher expression observed in the crypts than the villi of the small intestine (Figs. 2D, E and S2C, D). Positional scoring of *Rac1b*-positive signal along the crypt villus axis confirmed *Rac1b* expression was found primarily in the crypt region (Fig. S2E). However, expression was not restricted to the crypt base suggesting *Rac1b* expression is enriched in proliferative crypt cells but is not a specific marker of Lgr5+ stem cells (Fig. S2E). Tissue from induced *VilCre*[ERT2] *Rac1b*[fl/fl] mice (described later) was completely lacking hybridisation signal demonstrating the specificity of the BaseScope probe (Fig. S2C). To determine the effects of constitutive WNT signalling activation on *Rac1b* expression, we deleted *Apc* from the mouse small and large intestine (by tamoxifen induction of *VilCre*[ERT2] *Apc*[fl/fl] mice and analysis

5 days later) and stained for *Rac1b*. *Rac1b* signal was increased in crypts and villi in both tissues indicating increased *Rac1b* expression following *Apc* deletion (Figs. 2D–F and S2C, D). Immunohistochemical analysis of β-catenin indicated increased *Rac1b* expression in the villus of *Apc*-deleted intestines correlated with nuclear accumulation of β-catenin (Fig. S2F). Again, tissue from induced *VilCre*[ERT2] *Apc*[fl/fl] *Rac1b*[fl/fl] mice was almost completely negative for hybridisation signal (Fig. S2C). We also found significant overexpression of *Rac1b* following *Apc* deletion using qRT-PCR (Fig. S2G). These data demonstrate that *Rac1b* expression is increased following *Apc* deletion and correlates with WNT signalling activation, consistent with data from human tumours.

***Rac1b* deletion attenuates intestinal tumorigenesis and tumorigenic WNT signalling.** To test the functional significance of *Rac1b* expression in intestinal tumorigenesis we generated mice carrying a novel floxed allele permitting deletion of *Rac1* exon 4 (*Rac1b*[fl]) (Figs. 3A and S3A) and bred these mice onto the tamoxifen-inducible *VilCre*[ERT2] *Apc*[fl/+] intestinal cancer model[31]. In this model, tamoxifen induction leads to intestinal specific loss of a single copy of the *Apc* gene. When aged, stochastic loss of the WT *Apc* allele leads to tumour formation in the small and large intestines. We generated cohorts of *VilCre*[ERT2] *Apc*[fl/+] (*Apc*) and *VilCre*[ERT2] *Apc*[fl/+] *Rac1b*[fl/fl] (*Apc Rac1b*) mice, induced them with tamoxifen and aged them until clinical signs of intestinal tumorigenesis became apparent. Deletion of *Rac1b* significantly increased tumour-free survival compared to WT controls (median survival 138 vs 171 days) (Fig. 3B). Macroscopic and histological analysis of the intestines of these mice also revealed a significant reduction in average tumour number (54 vs 27) (Fig. 3C–E). Analysis in different regions of the intestine revealed a significant reduction in tumour number in the duodenum and jejunum, but not in the ileum or colon (Fig. S3B). qRT-PCR analysis demonstrated a significant increase in *Rac1b* expression in small intestinal tumours arising in *Apc* mice confirming our observations that WNT signalling activation increases *Rac1b* expression (Fig. S3C). Interestingly, analysis of tumours harvested from *Apc Rac1b* mice did not show the expected reduction in *Rac1b* expression, and analysis of individual tumours indicated around half of small intestinal and all colonic tumours expressed higher levels of *Rac1b* than normal intestinal tissue (Fig. S3C, D). This indicated a failure to efficiently delete *Rac1b* suggesting a positive selective pressure for maintaining *Rac1b* expression in tumour-initiating cells. Phenotypically, tumours from *Apc* and *Apc Rac1b* mice had comparable histological appearance with similar levels of apoptosis (as measured by Caspase 3

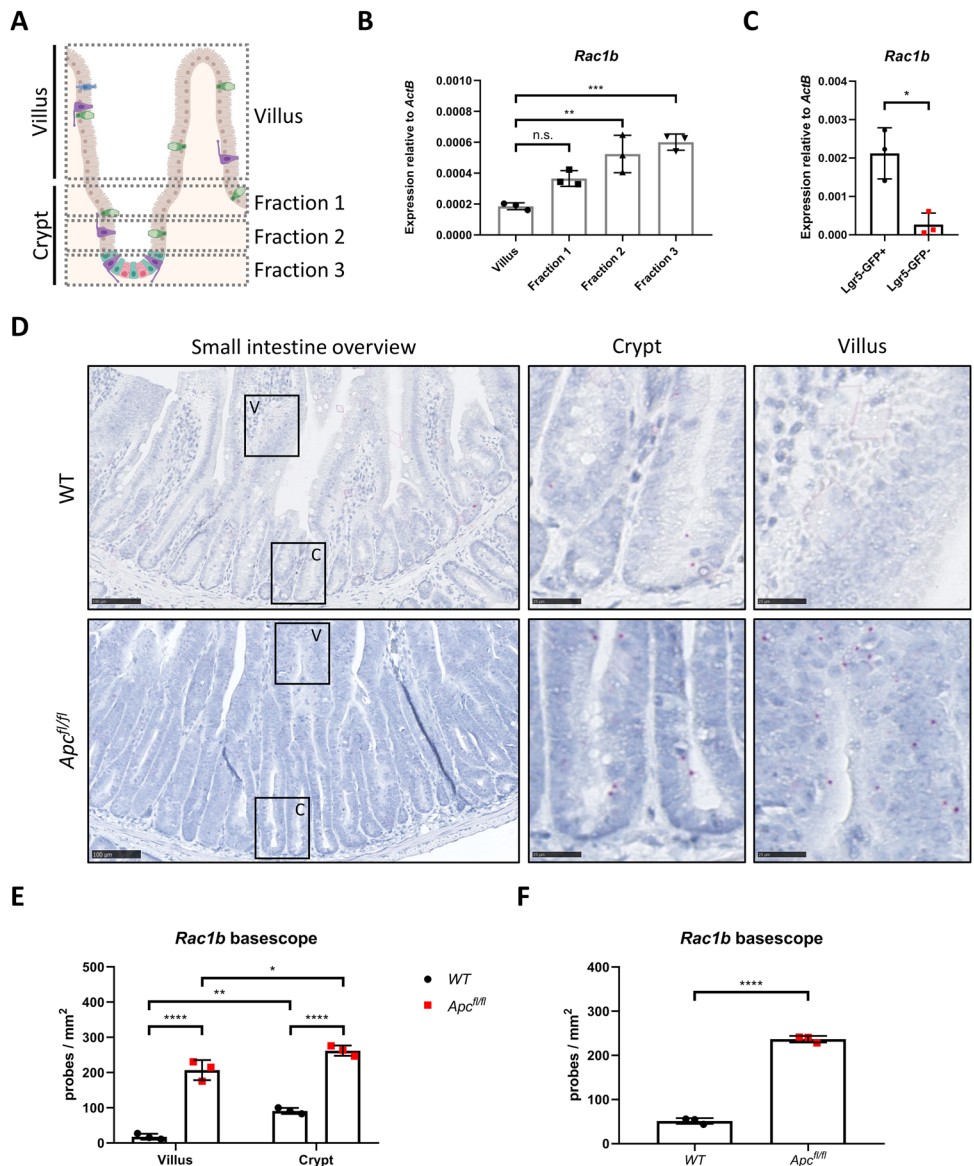

**Fig. 2 Rac1b is expressed in intestinal crypt cells and its expression increases following Apc deletion. A** Schematic outlining crypt extraction protocol. **B** qRT-PCR of Rac1b from villus and crypt fractions (data are presented as mean ± SD; $^{**}P = 0.0018$, $^{***}P = 0.0005$; one-way ANOVA with Tukey multiple correction, $n = 3$ biologically independent mice). **C** qRT-PCR of Rac1b from sorted Lgr5+ and Lgr5- cells (data are presented as mean ± SD; $^{*}P = 0.0117$; two-tailed t-test, $n = 3$ biologically independent mice). **D** Representative images of Rac1b basescope of WT and Apc$^{fl/fl}$ small intestinal tissue. Magnified areas are shown in right panels. Pink dots are positive probe detection. Scale bars are 100 μm (left panels) and 25 μm (magnified). **E** Quantification of Rac1b basescope probe counts comparing WT vs Apc$^{fl/fl}$ small intestine separated by villus and crypt regions (data are presented as mean ± SD; $^{*}P = 0.018$, $^{**}P = 0.0034$, $^{****}P = 4 \times 10^{-6}$ (WT vs Apc$^{fl/fl}$ villus), $^{****}P = 8.62 \times 10^{-6}$ WT vs Apc$^{fl/fl}$ crypt); two-way ANOVA with Tukey multiple correction, $n = 3v3$ biologically independent mice. For each mouse, at least five areas for crypt and villus regions were scored. In total, these areas incorporated at least 50 crypt/villus axes and covered ~2 mm$^2$ total area). **F** Quantification of Rac1b basescope probe counts comparing WT vs Apc$^{fl/fl}$ across the entire small intestine (data are presented as mean ± SD; $^{****}P = 5.47 \times 10^{-6}$; two-tailed t-test, $n = 3v3$ biologically independent mice). Source data are provided as a Source Data file.

immunohistochemistry, IHC) and similar numbers of Paneth and Goblet cells (Fig. S3G, H). However, tumours from Apc Rac1b mice were significantly less proliferative than controls (Fig. 3F, G). To determine whether there is a direct association between RAC1B and tumour-cell proliferation, we used Rac1b Basescope to identify tumours from our Apc Rac1b mice that had not efficiently deleted Rac1b (Fig. S3E, F). By staining these tumours for BrdU incorporation, we found that these tumours proliferate significantly better than those that are Rac1b-negative and to a similar extent as tumours from Apc control mice (Fig. S3E, F). Thus, Rac1b expression is required for efficient tumour-cell proliferation. Importantly, Rac1b deletion in normal adjacent

tissue was maintained for the duration of the experiment (~6 months) in all mice analysed (Fig. S3C, D), and cell proliferation, apoptosis, Paneth cell and goblet cell populations were unchanged suggesting Rac1b deletion is well tolerated in normal intestinal tissue (Fig. S3I, J). In addition, analysis of Rac1 expression indicated this was mostly unaffected by deletion of Rac1b exon 4 demonstrating the specificity of our knockout allele (Fig. S3K). To further investigate the tumorigenic phenotype, we utilised an ex vivo organoid clonogenic culture method to determine the tumour-initiating capacity of Apc-deficient cells in the absence of Rac1b expression (Fig. S3L, M). We generated intestinal organoid cultures from mice following acute

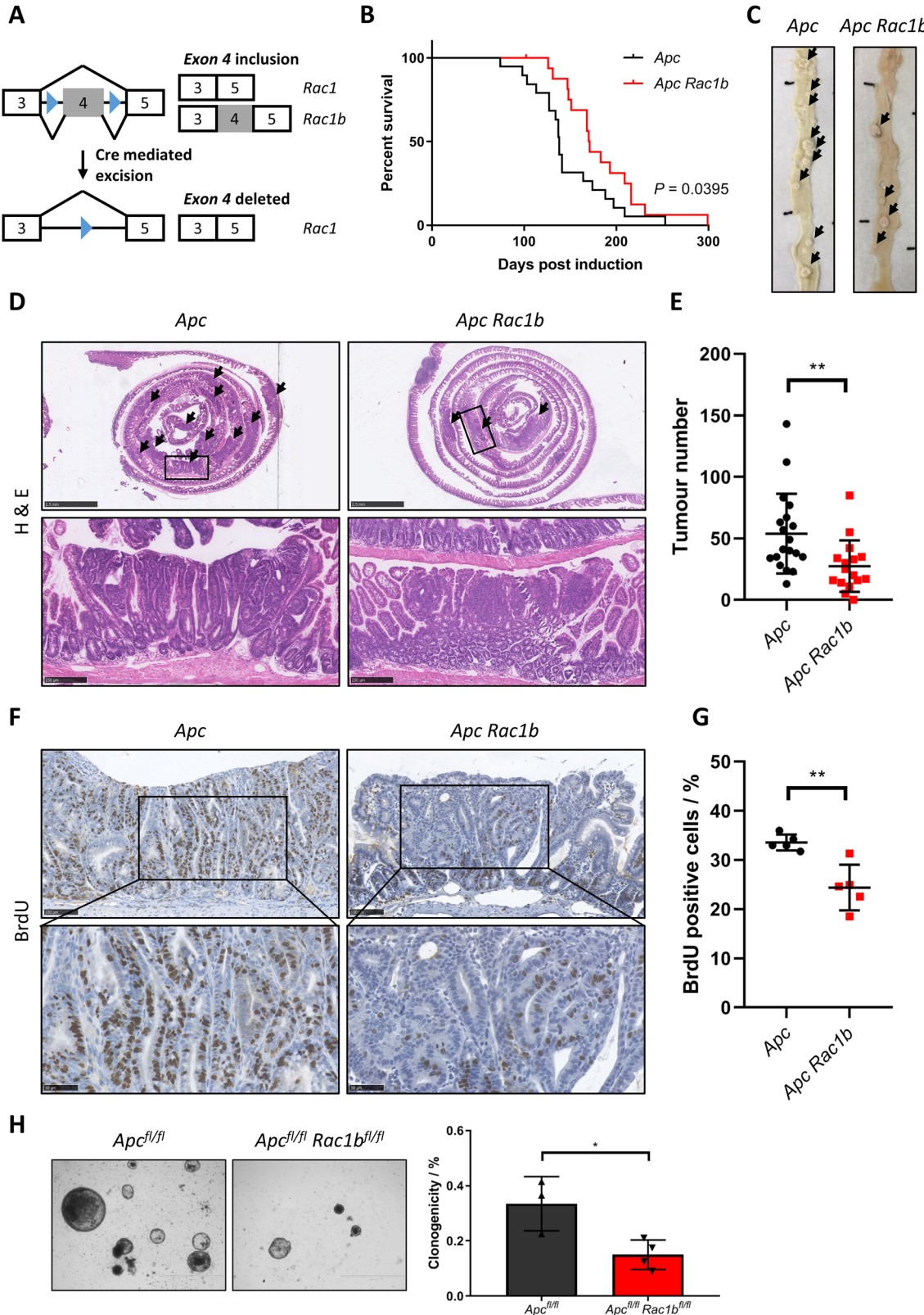

homozygous deletion of *Apc* and *Rac1b* (*Apc^fl/fl* and *Apc^fl/fl* *Rac1b^fl/fl*), digested them to single cells and determined their clonogenic capacity. Interestingly, the ability of *Apc*-deficient cells to form colonies from single cells was significantly reduced by *Rac1b* deletion (Fig. 3H). Together, these data demonstrate an important role for *Rac1b* expression during WNT-mediated intestinal adenoma formation.

To identify potential pathways mediated by *Rac1b* expression involved in tumorigenesis, we carried out RNAseq analysis on 4 vs 4 small intestinal tumours dissected from *Apc* and *Apc Rac1b* mice (confirmed negative for *Rac1b* by qRT-PCR). This analysis identified 506 mRNAs with significantly altered expression (fold change > 1.5, padj < 0.05). Of these, 265 had significantly increased expression and 241 had significantly decreased

**Fig. 3 Deletion of *Rac1b* suppresses intestinal tumorigenesis and tumour-cell proliferation. A** Schematic of *Rac1b* floxed allele. **B** Survival plot of *Apc* vs *Apc Rac1b* mice. Log-rank *P* value shown, *n* = 19v17 biologically independent mice. **C** Representative photographs of intestines from *Apc* and *Apc Rac1b* tumour-bearing mice. Black arrows indicate tumours. **D** H&E of whole intestinal rolls from *Apc* and *Apc Rac1b* mice. Box indicates magnified area. Black arrows indicate tumours. Scale bars are 2.5 mm (top panels) and 250 μm (magnified). **E** Quantification of total tumour number in the intestines of *Apc* vs *Apc Rac1b* mice (data are presented as mean ± SD; **$P$ = 0.003; two-tailed Mann-Whitney test, *n* = 19v16 biologically independent mice). **F** BrdU IHC in tumours from *Apc* vs *Apc Rac1b* mice. Scale bars are 100 μm (top panels) and 50 μm (magnified). **G** Quantification of BrdU positivity in tumours from *Apc* vs *Apc Rac1b* mice (data are presented as mean ± SD; **$P$ = 0.0079; two-tailed Mann-Whitney test, *n* = 5v5 biologically independent mice). **H** Clonogenicity assays of *Apc*[fl/fl] vs *Apc*[fl/fl] *Rac1b*[fl/fl] organoids (data are presented as mean ± SD; *$P$ = 0.0228; two-tailed *t*-test, *n* = 3v4 biologically independent mice). Scale bars are 1000 μm. Source data are provided as a Source Data file.

expression (Supplementary Data 1). Pathway analysis of our dataset identified regulation of immune response, antigen processing and viral response amongst the most upregulated pathways indicating a role for *Rac1b* in regulating these processes (Fig. 4A). Interestingly, amongst the most significantly down-regulated pathways were the canonical WNT signalling pathway and WNT-protein binding (Fig. 4A). Furthermore, GSEA demonstrated significant enrichment of genes downregulated in our RNAseq dataset with gene sets associated with WNT signalling activation (Fig. 4B, C). We used qRT-PCR to validate the differential gene expression of some of the identified stem cell/ WNT target genes in *Rac1b* deficient (Fig. 4D). There was no significant expression changes in any of these target genes in normal tissue from the same mice suggesting the requirement for *Rac1b* is specific to tumour cells. Intestinal tumours arising in *Apc Rac1b* mice demonstrated strong, nuclear β-catenin positivity suggesting a role for RAC1B downstream of β-catenin nuclear localisation (Fig. S4A). This is in agreement with our previous findings that β-catenin operates upstream of RAC1 signalling in promoting intestinal tumorigenesis[19]. Alongside our analysis of human TCGA data, these findings suggest a key role for RAC1B in regulating tumorigenic WNT signalling in CRC.

**RAC1B interacts with core components of the EGFR signalling pathway.** To better understand the mechanism by which RAC1B mediates intestinal tumorigenesis, we sought to identify its interacting proteins using proximity-dependent biotin identification (BioID) (Fig. 5A). We overexpressed RAC1 and RAC1B proteins fused to the BirA ligase (and BirA-only controls) in the mouse rectal tumour cell line CMT93, treated with biotin followed by streptavidin capture and identified enriched proteins by mass spectrometry. Here, 60 enriched proteins were identified (Supplementary Data 2) with both RAC1 and RAC1B having highly similar interactomes (50 common interactors, and only 2 specific to RAC1 and 8 specific to RAC1B) (Fig. 5B). Pathway analysis of identified proteins demonstrated enrichment in adherens junctions, focal adhesions and localisation to the plasma membrane (Fig. S5A). We used NetworkAnalyst and the IMEx interactome database to visualise the protein–protein interactions of the identified proteins. This analysis identified 7 protein sub-networks with the major network clustered around membrane localised components of oncogenic signalling pathways (Fig. 5C). This network contained core members of the EGFR/ERBB (EGFR, ERBB2), IGF (IGF1R), SRC (YES1) and EPH (EFNB2) signalling pathways. Further analysis of identified proteins indicated other EGFR-related signalling components such as ASAP1 and PTPRJ[32,33]. A number of proteins with links to the WNT signalling pathway were also identified, such as CCNY, which activates WNT signal through LRP6 receptor phosphorylation[34], and PROM1, a marker of intestinal stem/progenitor cells and colon cancer stem cells[35–37]. However, no members of the core receptor and/or destruction complex were identified indicating that, in our model, RAC1B may not function directly in the WNT signalling pathway. Interestingly, several reports have highlighted

potential crosstalk between EGFR and WNT signalling pathways[38,39]. Additionally, the EGFR signalling pathway is a therapeutic target in CRC, so the involvement of RAC1B on its activity may have implications for the treatment of this disease. Therefore, we chose to investigate the relevance of the identified RAC1B–EGFR pathway interaction. We first validated the interaction between RAC1B and EGFR. The BioID experiment was repeated by treating BirA- and BirA-RAC1B-transfected CMT93 cells with biotin followed by streptavidin-dependent pull-down. Western blot analysis of the pull-down fractions confirmed efficient, specific biotinylation of EGFR by BirA-RAC1B and not BirA control (Fig. S5B). Thus, EGFR and RAC1B are either interacting or neighbouring proteins in CRC cells.

**RAC1B is required for efficient stimulation of EGFR signalling.** We next determined whether RAC1B is required for EGFR and downstream signalling. We designed an antisense oligonucleotide to bind the putative exon splice enhancer of *Rac1* exon 4 to disrupt the inclusion of this exon. Vivo-Morpholinos targeting exon 4 (*Rac1b* PMO) demonstrated efficient in vitro knockdown efficacy with ~95% reduction in *Rac1b* transcript compared to treatment with a non-targeting, control Vivo-Morpholino (NT PMO) (Fig. S5C). Total *Rac1* transcript levels were not significantly altered following this treatment indicating specific inhibition of exon 4 inclusion without knockdown of total transcript (Fig. S5C). To determine the effects of *Rac1b* depletion on EGFR signalling, CMT93 cells were pre-treated with NT or *Rac1b* PMO for 2 days and then treated with 20 ng/ml EGF after serum starvation. Samples were collected for analysis 5, 10, 15 and 30 min after treatment and untreated cells grown in 10% serum or following starvation were used as controls. In control cells p-EGFR levels rapidly increased, reaching a peak at 10 min post treatment. *Rac1b*-deleted cells presented a reduction in the phosphorylated status of EGFR, implying lower levels of activation (Fig. 5D, E). This difference in p-EGFR was most apparent 10 min post stimulation and *Rac1b* PMO-treated cells never reached a level of phosphorylation as high as the NT PMO. To determine whether decreased EGFR phosphorylation translated into lower downstream EGFR pathway activation, we assessed ERK1/2 and AKT phosphorylation and found both proteins were less phosphorylated in *Rac1b* PMO-treated cells compared to controls (Fig. S5D, E). Replicate experiments performed in serum-starved cells with EGF treatment for 10 min confirmed these findings (Fig. S5F, G). Next, we sought to determine whether *Rac1b* regulates EGFR signalling in vivo in the *Rac1b*-deficient mice whose survival and tumorigenesis was diminished (Fig. 3). We analysed pEGFR and pERK levels in tumours derived from these mice and found that pEGFR levels are significantly lower in the *Rac1b*-deficient tumours but pERK levels are unchanged (Fig. S5H, I). This suggests that as tumours establish and develop, EGFR phosphorylation remains dependent on Rac1b expression but alternative mechanisms to activate pERK can be utilised. We also investigated the expression of known EGFR pathway target genes (*Etv4* and *Etv5*). Consistent with the

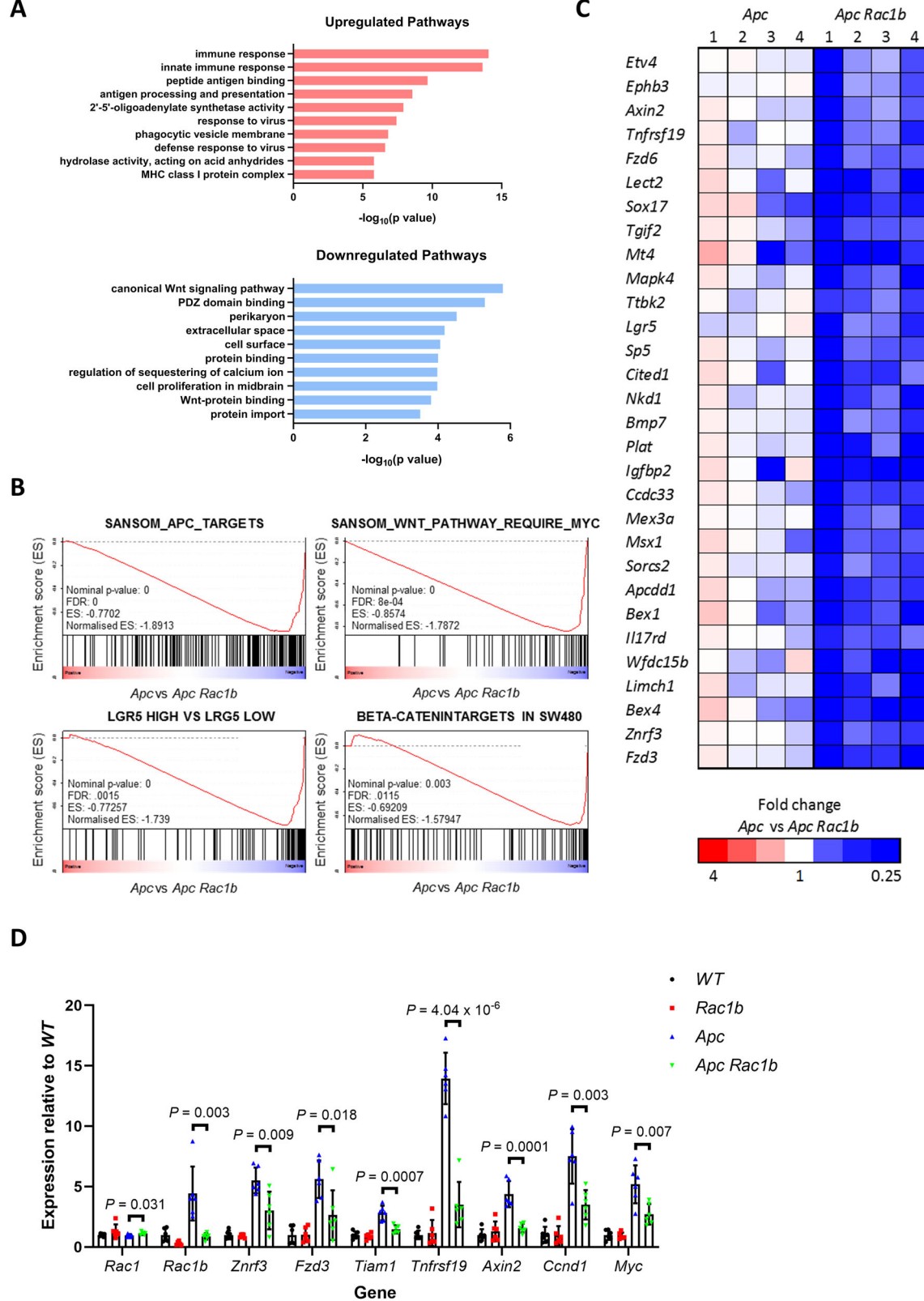

**Fig. 4 Oncogenic WNT signalling is mediated by RAC1B. A** Bar plot showing the top 10 significantly enriched upregulated and downregulated pathways in tumours from *Apc* vs *Apc Rac1b* mice. **B** GSEA plots of WNT target gene sets comparing tumours from *Apc* vs *Apc Rac1b* mice. **C** Heatmap of RNAseq expression values of WNT target genes and intestinal stem cell markers comparing tumours from *Apc* vs *Apc Rac1b* mice. **D** qRT-PCR analysis of various WNT target genes comparing tumours from *Apc* vs *Apc Rac1b* mice. As controls, matched normal tissue from these mice (*WT* and *Rac1b*, respectively) are also included (data are presented as mean ± SD; two-tailed *t*-test; *n* = 6v6 mice). Source data are provided as a Source Data file.

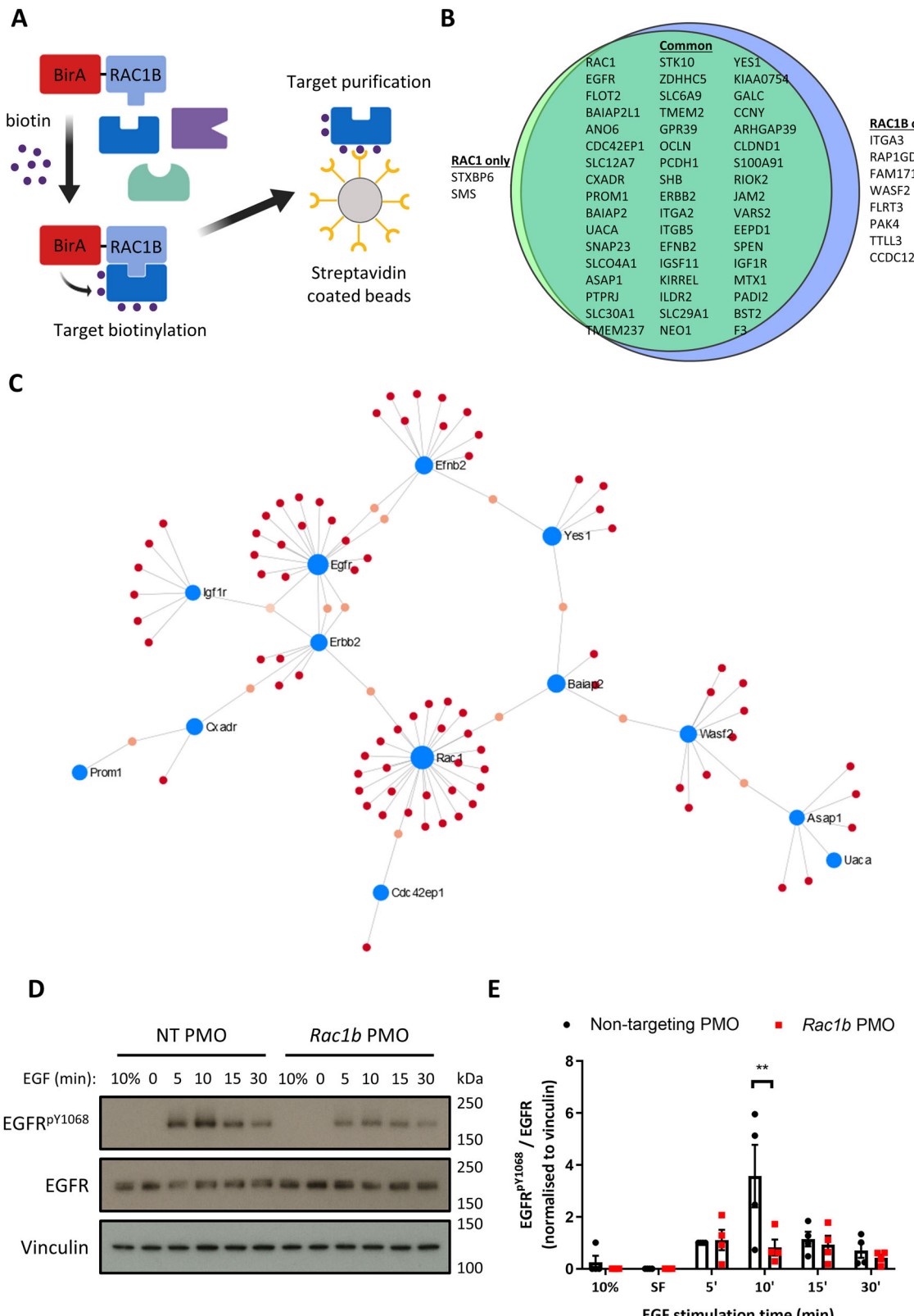

**Fig. 5 RAC1B interacts with multiple membrane-bound receptor tyrosine kinases. A** Schematic of BioID protocol. **B** Venn diagram outlining RAC1 and RAC1B interacting proteins identified by MS and the overlap between the interactomes. **C** Network analysis of the RAC1/RAC1B interactome. **D** CMT93 cells were treated with NT or *Rac1b* PMO, serum-starved and then stimulated with EGF for 5, 10, 15 or 30 min before analysis by western blotting for EGFR^pY1068, EGFR and Vinculin (sample control blot). Representative blots shown. **E** EGFR^pY1068 band densitometry relative to EGFR (normalised to Vinculin) of 4 replicate time course experiments (data are presented as mean ± SEM; **$P = 0.002$; two-way ANOVA with Tukey multiple correction; $n = 4v4$). Source data are provided as a Source Data file.

pEGFR reduction, expression of both is significantly lower in *Rac1b* deficient tumours (Fig. S5J).

To provide mechanistic insight into how RAC1B controls EGFR signalling we investigated EGFR trafficking using Alexa 555-labelled EGF (EGF-555). Confocal analysis of EGF-555-stimulated cells indicated that internalisation of EGF-555 was not significantly impaired in *Rac1b*-depleted cells (Fig. S6A). However, the localisation of internalised EGF-555 appeared altered. In control cells, we observed EGF-555 foci throughout the cell, whereas in *Rac1b*-depleted cells we observed them around the cell periphery (Fig. S6B, inset panels). To investigate this in more detail, we used live cell imaging to monitor overall endocytic trafficking of EGFR using EGF-555. Tracking the paths of EGF-555 vesicles revealed that the dynamic trafficking pattern in control cells was restricted following *Rac1b* depletion (Fig. S6A, C). We found that both total EGF-555 displacement and tracking velocity were significantly lower in *Rac1b* depleted cells suggesting altered EGFR trafficking (Fig. 6B, C). To investigate this further, we assessed the co-localisation between EGFR and the early endosome marker Rab5. In agreement with our previous observations that EGF-555 internalisation was unaffected by *Rac1b* depletion, EGFR co-localisation with Rab5 endosomes following EGF-555 stimulation was unchanged in *Rac1b* depleted cells (Fig. S6D). Together, this suggests *Rac1b* depletion does not prevent EGF internalisation into early endosomes but leads to an impairment of subcellular EGFR trafficking. Following internalisation, EGFR can be either recycled back to the cell surface to maintain signalling or directed towards multivesicular bodies for lysosomal targeting[40]. To investigate whether the absence of RAC1B promotes a shift in EGF-bound EGFR destination fate, NT and *Rac1b* PMO cells were stimulated with EGF-555 and treated with LysoTracker Green dye to identify lysosomal compartments. In control cells, co-localisation of labelled EGF and LysoTracker was observed from 30 min post EGF stimulation. However, *Rac1b*-depleted cells present significantly more EGF-555 Lysotracker clusters starting at earlier time points suggesting that deletion of *Rac1b* promotes EGFR lysosomal sorting and receptor degradation over the recycling route (Fig. 6D, E). Together, these data suggest RAC1B is required for efficient trafficking and recycling of EGFR, thus potentiating its activity.

We next investigated how RAC1B modulates WNT signalling, in particular the potential link between these signalling pathways in intestinal cancer. Firstly, we investigated the impact of EGFR inhibition on WNT target gene expression in *Apc*-deficient organoids and found that, similar to following *Rac1b* deletion, a subset of targets were downregulated (Fig. S6F). In addition, we depleted *Rac1b* in *Apc*-deficient organoids carrying an additional activating mutation of *Kras*. In these organoids, where MAPK signalling is constitutively active, downstream of EGFR, the depletion of *Rac1b* had no effect on organoid clonogenic capacity or WNT target gene expression (Fig. S6G–J). We also investigated the possible involvement of NF-κB signalling and ROS activity but found these were unaltered in *Rac1b* deficient organoids (Fig. S6K, L). Interestingly, we have recently shown that deletion of a number of RacGEFs leads to reduced ROS activity in *Apc*-deficient tissue, suggesting that RacGEF-activated RAC1 controls ROS production but RAC1B does not (Pickering et al.[41]). Together, these data suggest there is crosstalk between WNT and EGFR signalling pathways in intestinal cancer and RAC1B mediated control of EGFR signalling activation is required for efficient WNT signalling activity.

**RAC1B depletion sensitises EGFR inhibitor-resistant colorectal cancer liver metastatic organoids to cetuximab treatment.** These results suggest that RAC1B modulates EGFR receptor and signalling activation. To determine whether RAC1B stimulation

of EGFR signalling was an important mediator of tumorigenic growth, we grew *Apc*^fl/fl^ and *Apc*^fl/fl^ *Rac1b*^fl/fl^ intestinal organoids in the presence or absence of EGF ligand. Whereas in the presence of EGF organoids from both lines grew to a comparable size, in its absence the growth of those lacking RAC1B was significantly perturbed (Fig. S7A). We found similar results when the same organoid lines were treated with the EGFR inhibitor PD153035, with *Apc*^fl/fl^ *Rac1b*^fl/fl^ organoids being significantly more sensitive to this treatment (Figs. 7A, B and S7B). To determine whether inhibition of RAC1B might cooperate with EGFR inhibition to increase the efficacy of treatment in human tumour organoids, we designed a Vivo-Morpholino to target human *RAC1* exon 4 (*hRAC1B* PMO) (Fig. 7C). We cultured patient-derived organoids (PDOs) from a benign colonic polyp and a stage 3 invasive tumour (T2N1M0) from a familial adenomatous polyposis (FAP) patient undergoing tumour resection. PDOs treated with *hRAC1B* PMO demonstrated a robust (<95%) knockdown in *RAC1B* transcript and protein levels (Figs. 7D and S7C). As before, total *RAC1* transcript was not significantly altered (Fig. 7D). To determine the effects of *RAC1B* knockdown on human tumour organoid growth, we pre-treated PDOs derived from the benign colonic polyp with *hRAC1B* or NT PMO, digested to single cells and carried out clonogenicity experiments in the continued presence of morpholino. In agreement with our mouse in vivo and in vitro data, *RAC1B* depletion led to a significant reduction in clonogenic capacity (Fig. S7D). Interestingly, organoids derived from the more advanced invasive tumour resected from the same patient did not show reduced clonogenicity suggesting depletion of *RAC1B* alone is unable to suppress proliferation of later stage, more aggressive tumours (Fig. S7E). However, removal of EGF ligand or addition of EGFR inhibitor led to a significant reduction in growth of tumour organoids when combined with *RAC1B* knockdown (Fig. S7E). Collectively, these results indicate a functional relevance for RAC1B-dependent EGFR regulation and suggest a potential beneficial therapeutic effect when both are simultaneously inhibited. To address this in a more clinically relevant model, we obtained previously described cetuximab-resistant human CRC liver metastatic organoids[42]. Two independent organoid lines were tested, C001 and C002, both are *KRAS* WT, both express high levels of *RAC1B* compared to the stage 3 invasive tumour utilised above (Fig. S7F) and both show resistance to cetuximab treatment in vitro. In addition, C002 was derived from a patient who acquired resistance to cetuximab upon treatment. In line with previous reports, treatment of both organoid lines with cetuximab did not affect organoid growth (Figs. 7E, F and S7G–J)[42]. However, combining cetuximab treatment with *RAC1B* depletion led to a significant inhibition of organoid growth (Figs. 7E, F and S7G–J). These results suggest the cooperation between RAC1B and EGFR signalling we identified in our mouse models is conserved in human CRC, and depletion of RAC1B can enhance the potency of EGFR inhibition in cetuximab-resistant tumours.

## Discussion

Here, we demonstrate that RAC1B mediates intestinal tumorigenesis. Using genetic mouse models we show that RAC1B is required for efficient formation of intestinal tumours with deletion of *Rac1b* leading to a reduction in tumour number, reduced tumour proliferation and decreased oncogenic WNT signalling. We find that RAC1B interacts with EGFR, and controls the activation of the EGFR signalling pathway by mediating its intracellular trafficking providing a plausible mechanism for its pro-tumorigenic functions. Using a novel inhibitor of *RAC1B* splicing, we demonstrate enhanced sensitivity to EGFR inhibition

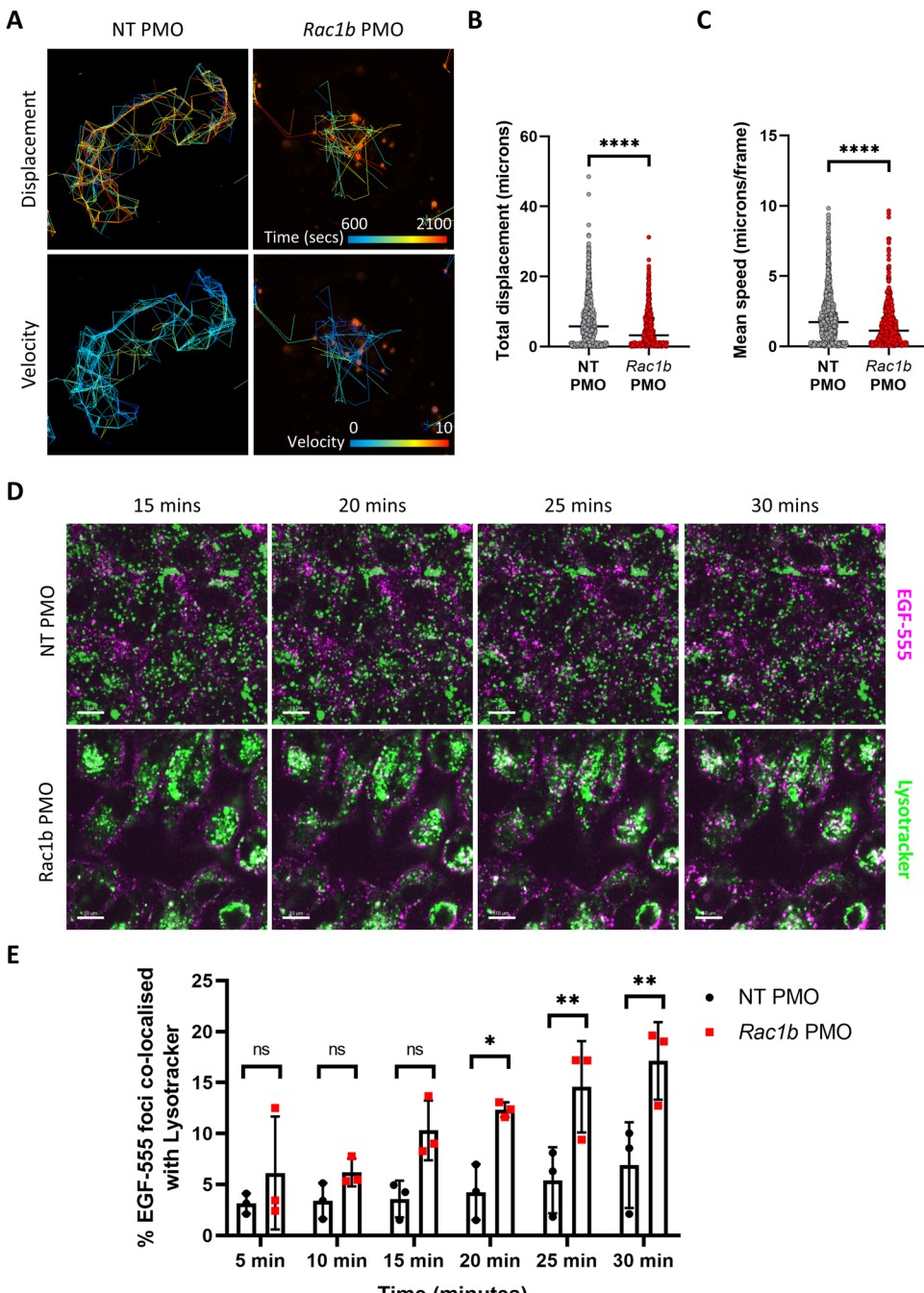

**Fig. 6 RAC1B mediates EGFR signalling and its depletion sensitises mouse tumour organoids to EGFR inhibition. A** Live cell imaging of Alexa 555-EGF tracking with time and velocity represented as colour spectrums. Tracking started after addition of EGF-555 and was carried out for the indicated time period. **B** Individual value plot of quantification of tracking distance of EGF-555-positive vesicles (median value indicated; [****]$P = 1.67 \times 10^{-13}$; two-tailed $t$-test; $n = 1213$ vs 658 foci). **C** Individual value plot of quantification of tracking velocity of EGF-555-positive vesicles (median value indicated; [****] $P = 6.93 \times 10^{-11}$; two-tailed $t$-test; $n = 1213$ vs 658 foci). **D** Representative images of live cell imaging of Alexa EGF-555 (magenta) and LysoTracker Green (green) at indicated time points post EGF-555 addition. White arrows indicate areas of EGF-555 LysoTracker co-localisation. Co-localised foci are observed as white dots. Scale bars are 10 μm. **E** Quantification of the percentage of EGF-555 foci co-localised with LysoTracker Green at indicated time points (data are presented as mean ± SD; [*]$P = 0.0263$, [**]$P = 0.0094$ (25 min), [**]$P = 0.0094$ (30 min), two-way ANOVA with Tukey multiple correction; $n = 3$v$3$ independent treatment experiments). Source data are provided as a Source Data file.

following RAC1B depletion in cetuximab-resistant human tumour organoids. We also find that in human CRC, high levels of *RAC1B* associate with elevated WNT signalling and poor prognosis. Together, these findings highlight an important oncogenic function of *RAC1B* and suggest therapeutic targeting of RAC1B may enhance the efficacy of EGFR inhibition.

Previous findings from in vivo models have shown that *Rac1b* overexpression alone is insufficient to drive tumorigenesis in both intestine and lung[43,44], suggesting RAC1B function is important for tumour initiation and progression in cooperation with other oncogenic driver events. Our analysis of TCGA data is in agreement as it shows a significant association of high *RAC1B*

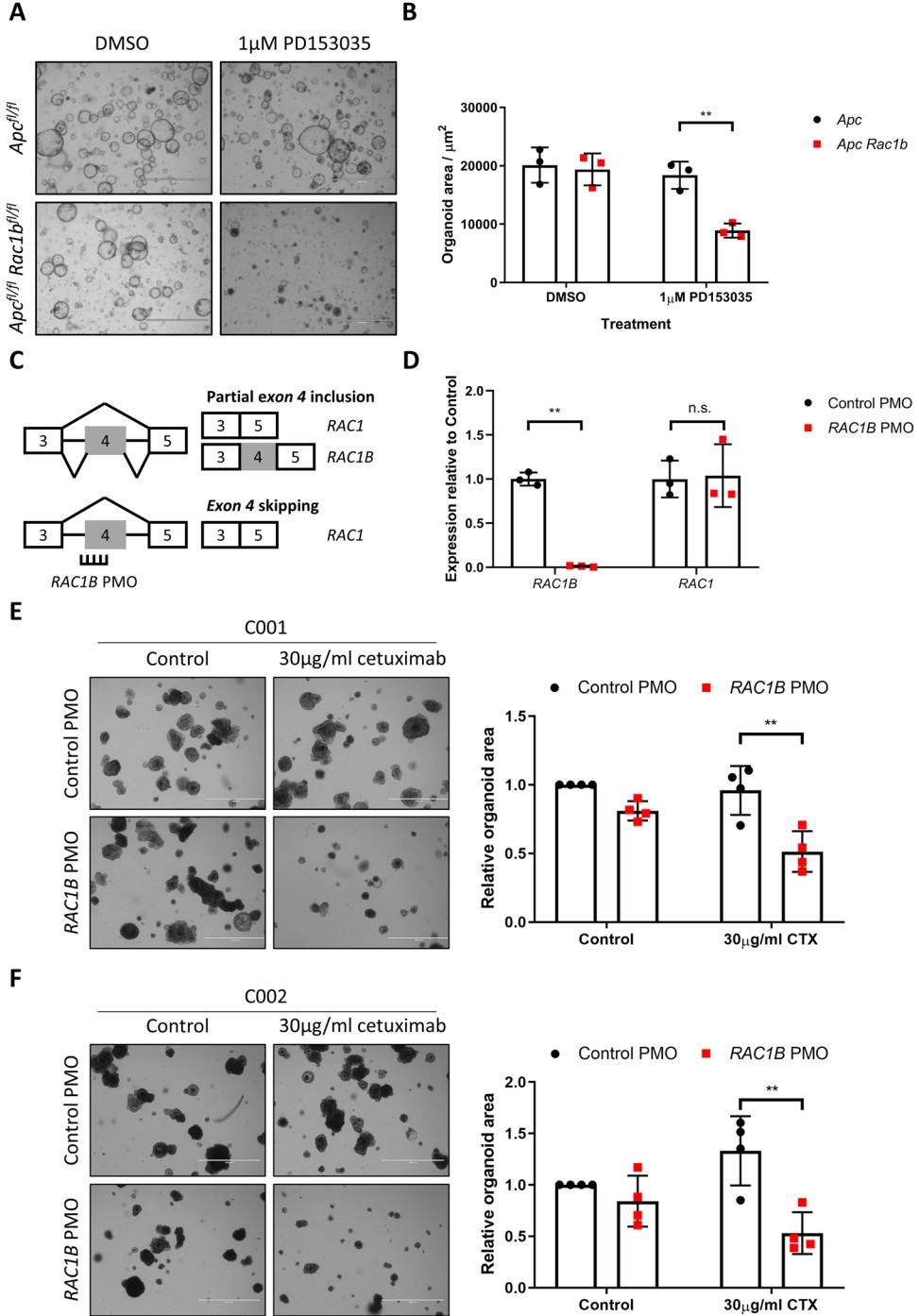

**Fig. 7 Depletion of *RAC1B* in human tumour organoids sensitises to EGFR inhibition. A** Representative images of *Apc*^fl/fl and *Apc*^fl/fl *Rac1b*^fl/fl organoids grown with or without the EGFR inhibitor PD153035. Scale bars are 1000 μm. **B** Quantification of organoid size 20 h after addition of PD153035 (data are presented as mean ± SD; **P = 0.0061; two-way ANOVA with Tukey multiple correction; n = organoids from 3v3 mice). **C** Schematic outlining design of *RAC1B* PMO. **D** qRT-PCR analysis of PDOs treated with control or *RAC1B* PMO (data are presented as mean ± SD; **P = 0.0019; two-way ANOVA with Tukey multiple correction; n = 3v3 independent treatment experiments). **E** Representative images (left panel) and quantification (right panel) of C001 organoid PDOs treated with control or *RAC1B* PMO in the presence or absence of cetuximab (left panel) (data are presented as mean ± SD; **P = 0.0011; two-way ANOVA with Tukey multiple correction; n = 4v4 independent treatment experiments). Scale bars are 1000 μm. **F** Representative images (left panel) and quantification (right panel) of C002 organoid PDOs treated with control or *RAC1B* PMO in the presence or absence of cetuximab (left panel) (data are presented as mean ± SD; **P = 0.0019; two-way ANOVA with Tukey multiple correction; n = 4v4 independent treatment experiments). Scale bars are 1000 μm. Source data are provided as a Source Data file.

expression with later tumour stage, presence of metastatic disease and poor prognosis. We also identified a significant correlation of elevated *RAC1B* and the WNT high, CMS2 CRC subtype and a negative correlation of *RAC1B* expression and the MSI high,

BRAF-mutated CMS1 subtype. Apparently contradictory to the survival data, CMS2 subtype CRC is characterised by relatively good prognosis[1]. We subsequently found that survival of patients with CMS2 subtype tumour could be separated based on *RAC1B*

expression, with the poorest prognosis in patients with high *RAC1B*. This subgroup of CMS2 patients perform as poorly as those with highly aggressive CMS4 subtype tumours suggesting a potential role for *RAC1B* in promoting tumorigenesis. It is possible that *RAC1B* is merely a marker of WNT-activated CRC, however, the suppression of WNT signalling in RAC1B-deficient tumours derived from our intestinal cancer model suggests that RAC1B is a functional mediator of WNT signalling. Another characteristic of CMS2 tumours is a favourable response to treatment with EGFR inhibitors. Interestingly, other CRC subtyping attempts have identified a WNT-activated subtype (termed TA) with relatively good prognosis[45]. However, the TA subtype contains two distinct populations that differ in their responsiveness to EGFR inhibitors termed TA-CS (cetuximab sensitive) and TA-CR (cetuximab resistant). Intriguingly, the TA-CR group displays high WNT activity and poor prognosis, similar to the *RAC1B* high, CMS2 group we have identified in this study. Therefore, it is tempting to speculate that high RAC1B expression plays a functional role in promoting the WNT high, CMS2 subtype and may promote characteristics of the cetuximab-resistant TA-CR subtype. If this is the case, inhibition of RAC1B could be of therapeutic benefit to patients with *RAC1B* high, CMS2 subtype tumours.

A number of previous studies have linked RAC1 and RAC1B to WNT signalling; so to define the mechanism by which RAC1B modulates tumorigenesis, we investigated its interactome using BioID. We found a number of proteins linked to WNT signalling, including CCNY and PROM1 but no core WNT receptor or signalling components were identified. The interactions with CCNY and PROM1 warrant future investigation but it is also possible that RAC1B modulates alternative pathways that cooperate with WNT signalling, to drive tumorigenesis. Of particular note was our identification of numerous RTKs as RAC1/RAC1B interacting proteins including the epidermal growth factor receptors EGFR and ERBB2. Both are well-defined drivers of oncogenesis, and inhibitors of EGFR are used clinically for the treatment of KRAS WT CRC. Interestingly, numerous reports have identified crosstalk between EGFR and WNT signalling. In particular, a recent study has shown that EGFR is required for WNT signalling via FZD9B in haematopoietic stem cells. This study shows that phosphorylation of FZD9B by EGFR promotes internalisation and signal transduction of the WNT9A–FZD9B–LRP complex providing evidence of a direct link between these pathways[38]. It is not clear if this mechanism is important for receptor-independent WNT activation such as following *Apc* loss, but our observations of changes to WNT target gene expression following EGFR inhibition suggest that links between these pathways are maintained under such conditions. In our study, we show that RAC1B depletion attenuates EGFR phosphorylation and downstream phosphorylation of AKT and ERK. Thus, efficient EGFR pathway activation requires RAC1B. Activation of EGFR signalling is controlled, in part, by the endocytic machinery, which directs receptors towards recycling or lysosomal sorting. Given the requirement for RAC1B to maintain EGFR phosphorylation after EGF stimulation, we speculated that RAC1B might interfere in EGFR intracellular dynamics. Indeed, using live cell imaging we observed altered EGFR trafficking upon *Rac1b* depletion and increased EGFR co-localisation with lysosomes. Therefore, it appears the depletion of *Rac1b* modifies EGFR trafficking, leading to increased lysosomal sorting and thus reduced activity. Our findings are in agreement with other studies that have demonstrated crosstalk between RAC1 and EGFR signalling pathways. For example, EGFR has been shown to activate RAC1 by accumulating the RacGEF TIAM1 at the membrane and promoting its GTP-bound state[46]. Additionally, EGF has been shown to promote RAC1B expression itself through modulation of the splicing factor HNRNPA1 although we did not find direct evidence of this

in our model[47]. Together with our data, this suggests a complex network of interacting signalling pathways where WNT signalling can promote RAC1B expression, which in turn promotes efficient activation of EGFR, thus driving tumorigenesis.

In this study, we have developed a novel method of modulating *RAC1B* splicing in vitro using an antisense morpholino. Antisense morpholinos have previously been utilised to induce exon skipping, with the approach showing promise as a therapeutic option for Duchenne muscular dystrophy and Spinal muscular atrophy[48]. Targeting *RAC1B* in this way is extremely specific and efficient, therefore, it may be a viable way of inhibiting the splicing of *RAC1B* without disrupting the function of *RAC1* itself. The effects of RAC1B depletion in human-derived tumour organoid cultures corresponded well with our in vivo findings. RAC1B depletion in a benign polyp was sufficient to significantly attenuate clonogenic capacity, but in organoid cultures derived from invasive and metastatic tumours it had little effect. This would likely rule out the possibility of targeting RAC1B as a single-agent therapeutic in late stage disease. However, treating RAC1B-depleted tumour cells with an EGFR inhibitor led to a significant reduction in the clonogenic growth of these cells. Importantly, we also found that depletion of RAC1B in cetuximab-resistant liver metastatic organoids sensitised them to cetuximab treatment. The EGFR inhibitors, cetuximab and gefitinib, are utilised for the treatment of late stage KRAS WT CRC. Although initial responses are observed in some patients, resistance rapidly emerges leading to disease progression. In some cases, resistance is driven by activation of MAPK pathway via mutation of genes such as KRAS, NRAS, NF1 or BRAF. In others, no obvious mutational drivers of resistance are present, suggesting non-genetic drivers of resistance. Intriguingly, a recent paper has suggested the acquisition of resistance to cetuximab can occur via the transcriptional switching from CMS2 to CMS4 subtypes[7]. CMS4 CRC is characterised by an EMT-like, TGF-β-activated transcriptome. We observed high *RAC1B* expression in a proportion of CMS4 tumours and it has previously been demonstrated that RAC1B can drive EMT, suggesting a potential functional role for RAC1B in this process[49]. In addition, activation of lncRNA MIR100HG-derived miR-100 and miR-125b have been observed in CRC cells that have acquired resistance to cetuximab[50]. These microRNAs promote WNT signalling activation via repression of multiple negative regulators of the pathway suggesting WNT signalling activation may be an important mechanism driving cetuximab resistance. Whilst our strategy of depleting *RAC1B* using morpholinos remains unproven in vivo, our proof-of-principle findings that depletion of RAC1B sensitises tumour organoids to EGFR inhibition in vitro suggest RAC1B may be considered as a potential therapeutic target for enhancing the efficacy of clinically used EGFR inhibitors.

## Methods

**Mouse experiments**. All mouse experiments were performed under the UK Home Office regulations and all relevant ethical regulations were adhered to. The study protocols were approved by University of Edinburgh AWERB. Mice were bred at the animal facilities of the University of Edinburgh and the Beatson Institute, were kept in 12 h light–dark cycles and were given access to water and food ad libitum. Mice were maintained in a temperature- (20–26 °C) and humidity- (30–70%) controlled environment. Mice (of both genders) were induced with tamoxifen between the age of 6 and 12 weeks, once they had reached a minimum weight of 20 g. Mice colonies had a mixed background (50% C57Bl6J, 50% S129). The genetic alleles used for this study were as follows: *vil-Cre-ER*^T2 [51], *Apc*^fl [52] and *Rac1b*^fl. Mice were genotyped by Transnetyx (Cordoba, USA). To achieve deletion of RAC1B protein without affecting RAC1 expression, loxP sites were inserted on either side of exon 4 of the mouse *Rac1b* transcript (Rac1-202; ENSMUST00000100489.3; Chromosome 5: 143,509,597 – 143,509,653). Details are given in "Supplementary Materials and Methods". For long-term Cre recombination in tumour cohorts, mice were induced with a single intraperitoneal (IP) injection of 80 mg/kg of tamoxifen and were aged until symptomatic of disease

(rectal bleeding, weight loss, hunched position and/or pale feet). For short-term Cre recombination, two consecutive IP doses of 120 mg/kg and 80 mg/kg of tamoxifen were administrated to the mice and they were sacrificed at day 5 post induction. Mice were randomly distributed by sex and age and a minimum of 15 or 3 mice were used for long- and short-term experiments, respectively. Proliferation was evaluated by a single IP injection of 200 µl of BrdU cell proliferation label (GE, 1392253) 2 h prior to termination. In long-term tumour cohort mice, tumour number and burden were macroscopically quantified in situ after mouse termination.

**Analysis of TCGA data.** For isoform expression analysis of the TCGA COAD dataset, normal and matched tumour tissue expression data of RAC1 (uc003spx.3) and RAC1B (uc003spw.3) isoforms were downloaded from the TCGA Synapse group (accession doi:10.7303/syn300013) and the IlluminaHiSeq_RNASeqV2 study. For RAC1B PSI analysis, data from each tumour type containing normal tissue expression were downloaded from the TCGASpliceSeq web-based platform[30]. RAC1B high and low groups were categorised according to whether PSI value was above 0.2 of normal tissue PSI average (high) or below the average of normal tissue PSI value (low). Information about patient survival, lymphovascular invasion and metastasis was extracted from CBioPortal (cbio-portal.org), as well as the mutational data of TCGA tumours and their CMS classification. For CMS and RAC1B correlation analysis, RSEM-calculated reads of TCGA COAD tumours' dataset were downloaded from Wang et al. study[53] and genes of each CMS subtype were correlated to its RAC1B PSI value. Moreover, the expression of some genes of interest was grouped according to RAC1B high or low tumours. Tumour stage data, MSI status and the presence of BRAF[V600E] mutation were acquired from the clinical data published by the CRC Subtyping Consortium (CRCSC) (accession doi: 10.7303/syn2623706).

**RNA sequencing and analysis.** Following RNA isolation using Qiagen RNeasy Mini Kit from intestinal tumours, RNA integrity was evaluated on an Agilent 2200 Bioanalyser and a RIN over 8 was considered optimal for analysis. Illumina RNAseq and library preparation were performed by Edinburgh Genomics (UK). Transcript abundances were quantified using kallisto[54] from a transcriptome index compiled from coding and non-coding cDNA sequences defined in GRCm38 (Ensembl 95). Differential expression was called using DESeq2[55] at the gene level after importing estimated counts per transcript from kallisto using tximport[56]. Significant genes were considered for pathway analyses using the Innate DB database (innatedb.com). Genes were pre-ranked and analysed for Gene Set Enrichment Analysis (GSEA, v3.0) to generate enrichment plots for Apc-, Wnt-, Lgr5- and β-catenin-related target genes' datasets.

**Organoid culture.** For organoid culture, intestinal epithelial cells were isolated as detailed in the 'Intestinal epithelial isolation and fractionation' section below. During cell fractionation step, fractions 3 and 4 were combined, washed with 20 ml of Advanced DMEM/F-12 medium (ADF, Gibco) supplemented with 100 U/ml penicillin, 100 µg/ml streptomycin, 2 mM L-Glutamine and 10 mM HEPES (Life Technologies, 15630080). This was centrifuged at 300*g* for 3 min, supernatant was discarded and the pellet was resuspended with 10 ml of supplemented ADF (from now on, ADF) and filtered through a 70 µm cell strainer to obtain isolated crypt fragments. Crypts were centrifuged at 700*g* for 10 min and pellet was carefully resuspended with 500 µl of Cultrex PathClear Reduced Growth Factor BME (Bio-Techne, 3533–010–02). In a pre-warmed 24-well plate, a volume of 20 µl/well of BME was plated. The plate was incubated for 10 min at 37 °C to allow BME solidification and 500 µl of growth medium was added. Growth medium was composed of ADF, 1X N2 (Gibco, 17502048), 1X B27 (Gibco, 17504044), 50 ng/ml EGF (Peprotech, 315–09–500) and 100 ng/ml Noggin (Peprotech, 250–38–500). PDOs were established and grown as previously described[57]. Ethical approval for human CRC organoid derivation was carried out under NHS Lothian Ethical Approval Scottish Colorectal Cancer Genetic Susceptibility Study 3 (SOCCS3) (REC reference: 11/SS/0109). Briefly, tumours were cut and washed with ADF in a 15 ml Falcon tube. Tumoral pieces were digested with 2 ml of digestion medium containing 1700 µl ADF, 1 mg/ml Collagenase II (Sigma, C1764), 0.5 mg/ml Hyaluronidase (Sigma, H3506) and 10 µM Y-27632 (Tocris, 1254). Digestion was performed at 37 °C for 90 min with vigorous pipetting every 15 min to facilitate digestion. Digestion was stopped by adding 100 µl of 1% BSA (bovine serum albumin). Cells were passed through a 40 µm cell strainer to obtain single cells and centrifuged for 3 min at 600*g*. Pellet was resuspended with 500 µl of BME and plated in a pre-warmed 24-well plate. PDOs were grown in ADF medium containing 100 U/ml penicillin, 100 µg/ml streptomycin, 2 mM L-Glutamine, 10 mM HEPES, 1X Primocin, 1X B27, 1.25 mM NAC, 10 mM Nicotinamide, 50 ng/ml EGF, 500 nM A83-01, 10 µM SB202190, 100 ng/ml Noggin, 1 µM PGE and 10 nM gastrin. The organoid cultures were passaged every 3–4 days.

**Depletion of Rac1b with an antisense morpholino.** Two antisense morpholinos were designed to specifically target the exon splice enhancer region of exon 4 in the mouse and human Rac1/RAC1 transcript:
    mRac1b PMO - GTCTATCTTTACCACATGTGTCTCC
    hRAC1B PMO - TCCTTACCGTACGTTTCTCCAACCT

Binding led to the skipping of the exon 4 and prevented translation of RAC1B protein. Morpholinos conjugated to an octa-guanidine dendrimer delivery moiety (Vivo-Morpholino, PMO) were purchased from GeneTools (USA) and a non-targeting morpholino (NT PMO) was used as control. For depletion of Rac1b in the CMT93 cells, $2.5 \times 10^5$ cells were seeded in a 6-well plate and the medium was supplemented with 2.5 µM of Rac1b PMO or NT PMO for 24 h. For depletion of RAC1B in human organoids, these were treated with 1 µM hRAC1B PMO or NT PMO for 96 h. Further experiments using 2D cells or 3D organoids deficient for RAC1B were conducted following this pre-treatment with morpholino. Morpholino was kept in the medium over the duration of the experiment.

**Clonogenicity assays.** Plated mouse organoids were washed and digested into single cells through incubation with 500 µl of StemPro Accutase solution (Life Technologies, A1110501) at 37 °C for 15 min. ADF was added to stop digestion and cells were filtered through a 40 µm cell strainer. Cells were spun down for 5 min at 600*g*, the pellet was resuspended with 2 ml of ADF and cells were automatedly counted using the Countess machine (Thermo Fisher). In all, 10,000 single cells were plated per 5 µl BME drop, and a minimum of 4 drops was plated per each genotype. Growth medium was added as usual and formed clones were counted after 4 days. The percentage of clonogenic capacity was calculated as the average percentage of spheres formed in each drop. This experiment was repeated with at least 3 biological replicates and statistical differences were calculated. For PDOs, organoids were digested with TripLE at 37 °C for 45 min with regular mixing. Single cells were filtered and counted as above and 2500 cells plated per 5 µl BME drop and a minimum of 4 drops was plated for each treatment group. Clone formation was scored 14 days later.

**Data plotting and statistical analysis.** All statistical analyses and graphs were performed using GraphPad Prism v7.0 software (La Jolla, USA) unless otherwise stated. Student's *t*-test was used for the comparison of two groups following normality. In contrast, groups that did not follow normality were analysed with the non-parametric Mann-Whitney test. Survival curves and percentage of survival were calculated with a Log-rank (Mantel-Cox) test. Pearson correlation was used to evaluate the correlation between two datasets, while Fisher's test was used to assess the association between two categories. All $P$ values less than 0.05 were considered statistically significant.

**Generation of Rac1b[fl] allele.** A conditional allele of Rac1b (Ensembl ID: ENSMUSG00000001847 in mouse genome assembly GRCm38.p6) was created by inserting loxP sites on either side of exon 4 (ENSMUSE00000648127; Chromosome 5: 143,509,597 - 143,509,653) of the Rac1b transcript (Rac1-202; ENSMUST00000 0100489.3). Small targeting vectors spanning the insertion sites of the loxP sites on the 5′ and 3′ side of Rac1b exon 4 were synthesised (GeneArt). An F3-Neo cassette and an Frt-Hygro cassette were inserted into the 5′ and 3′ loxP synthetic DNA plasmids, respectively, by co-transfection into EL250 Escherichia coli[58], which express Flp recombinase under arabinose induction. The loxP FRT Hygro DNA fragment bounded by the homology arms was excised and recombineered[58] into a mouse genomic DNA BAC clone (Source Biosciences) carrying the mouse Rac1b gene in EL250 E. coli[59]. The Hygro cassette was then removed by arabinose-induced Flp expression. Subsequently, the loxP F3 Neo DNA fragment was introduced into the same Rac1b BAC clone by recombineering. A linearised retrieval plasmid was generated by PCR of a p15A vector backbone with oligos incorporating 70 bp Rac1b homology arms[60]. The modified Rac1b sequences were retrieved from the BAC clone in EL250 E. coli by recombineering. The retrieved plasmid represents the targeting vector with approximately 6.5 and 3.4 kb homology arms. The targeting vector was linearised and transfected into HM1 mESCs[61]. Cells were selected under G418 (250 µg/ml) and surviving colonies picked and screened for targeting by long range PCR (using the Roche Expand Long Template PCR System) from within F3-Neo cassette to sequences beyond the ends of the homology arms. Oligo sequences used to screen cells to ensure appropriate targeting of the Rac1b gene were ATGTGGTATAGCTGTTCCCTGG TC and CTAGAGCTTGCGGAACCCTTAATG (7.3 kb) for the 5′ side and CTACTTCCATTTGTCACGTCCTGC and GTTGAGATGTGGTCCATGCTAA GC (5.3 kb) for the 3′ side. The presence of the isolated loxP site was confirmed by PCR with TTGGAGACACATGTGGTAAAGATAG and ACAGAACACCAGAG TCAGAGAAGAG (424 bp) and confirmed by digestion of the PCR product with XbaI, which is within the linked FRT site. Following identification of correctly targeted clones, mouse lines were derived by injection of ES cells into C57BL/6J blastocysts according to standard protocols. After breeding of chimeras, germline offspring were identified by coat colour and the presence of the modified allele was confirmed with the 3′ loxP primers described above. Mice were subsequently crossed with a mouse line expressing Flpe (Tg(ACTFLPe)9205Dym) to delete the selectable marker by recombination at the FRT sites[62]. Deletion with the selectable marker was confirmed by PCR across the remaining F3 site with the oligos CCCACAGATGAAACCAGGAG and GCTCAGCGTTCAGAAAGTGG (495 bp).

**Immunohistochemistry.** Intestinal tissue was harvested, flushed with PBS solution and fixed as "Swiss-roll" sections in PFA for 24 h at 4 °C. For tumour scoring, intestines were fixed in Methacarn (60% methanol, 30% chloroform and 10%

glacial acetic acid). Tissue was automatically processed through the Tissue-TeK VIP infiltration Processor (Sakura) for paraffin embedding and cut into 5 μm sections with the microtome (Leica). Standard IHC techniques were conducted during this study. Antibodies used were as follows: BrdU, 1:500 (Bioss, bs-0489H), β-catenin, 1:50 (BD Biosciences, 610154), cleaved Caspase 3, 1:800 (R&D), Lyz1 (DAKO, A009), Muc2 (Genetex, GTX100664), EGFR$^{PY1068}$, 1:25 (Cell Signalling, 3777S), EGFR$^{Y1068}$, 1:400 (Abcam, ab40815) and ERK1/2$^{pT202/Y204}$, 1:100 (Cell Signalling, 4370S). At least 3 different mice of each genotype were used as biological replicates in every IHC experiment. Scoring of the staining was done blinded for evaluation and representative images were selected. Images were digitalised using the Nano-zoomer Digital slide scanner (Hamamatsu) and analysed with the viewer software NDP.view2 (Hamamatsu). Scoring of tumour sections was automatically performed using the QuPath software (qupath.github.io), whilst proliferation scoring of normal intestine was carried out manually.

**BaseScope™ in situ hybridisation**. A specific BaseScope™ probe was designed against the exon 3–4 junction of the transcript variant 1 of *Rac1* (NM_001347530.1), which included the following nucleotides: GTTGGAGACACA TGTGGTAAAGATAGACCCTCCAGGGGCAAAGACAAGCCGATTGCCGA CGTGTTC. The probe was purchased from Advanced Cell Diagnostics (ACD Bio-Techne, UK) and BaseScope™ in situ hybridisation was conducted on formalin-fixed intestine tissues embedded in paraffin blocks following company instructions. For control of RNA integrity in tissue sections, sections were also hybridised with a reference positive-control probe. Three different mice per each genotype were used for the experiment and positive dots per tissue area were scored manually by blind scoring. For false colour analysis, to enhance visibility of Basescope dots, the image was split into red, green and blue channels in ImageJ. A threshold (180/255) was set in the blue image, and the colour channels remerged.

**Western blotting**. Proteins from organoids and cellular pellets were extracted using RIPA buffer (Sigma, R0278) supplemented with 1% of phosphatase and protease inhibitors (Sigma, P0044 and P8340). BCA Protein Assay kit (Pierce) was used to determine protein concentration. Here, 10 μg of denatured protein lysate was separated by electrophoresis in 4–12% Bis-Tris protein gels (NuPage, Thermo Fisher) and wet transferred onto methanol pre-activated nitrocellulose membrane. Quick staining with Ponceau solution was used to evaluate transfer efficiency and following PBST washes (PBS with 0.1% Tween, Sigma), membrane was blocked for 1 h at RT with 5% milk/PBST (dried milk, Marvel) or 3% BSA/PBST (BSA, Sigma) when probing phosphorylated proteins. The membrane was incubated with primary antibody diluted in the blocking solution o/n at 4 °C and with anti-rabbit or anti-mouse IgG (HRP linked, Cell Signalling) for 1 h at RT. Primary antibodies and concentrations used were as follows: AKT, 1:2000 (Cell Signalling, 9272S); AKT$^{P5473}$, 1:3000 (Cell Signalling, 4060S); EGFR, 1:1000 (Cell Signalling, 2232S); EGFR$^{PY1068}$, 1:500 (Cell Signalling, 2234S); ERK1/2, 1:2000 (Cell Signalling, 4695S); ERK1/2$^{pT202/Y204}$, 1:3000 (Cell Signalling, 4370S), NF-κB p65, 1:1000 (Abcam, ab7970), NF-κB p65$^{PS536}$, 1:1000 (Cell Signalling, 3033S), Rac1b (Millipore, 1:1000), Vinculin (Abcam, 1:5000), Streptavidin-HRP, 1:10,000 (Abcam, ab7403), Myc-tag, 1:10,000 (Cell Signalling, 2276S) and β-actin, 1:5000 (Cell Signalling, 4970S). Antibody signal was detected by chemiluminescence using the ECL Plus Substrate (Thermo Scientific) following product instructions, and the membrane was developed at the darkroom with ECL hyperfilm (Amersham). Bands' densitometry was assessed using FIJI (ImageJ). To allow subsequent use, membranes were stripped by incubation with a Stripping solution (Millipore, 2504) for 15 min at RT.

**RNA isolation**. Intestinal tissue and tumours preserved in RNAlater (Sigma, R0901) and pellets from tumour organoids or 2D cells were used to isolate RNA. RNA extraction was conducted using the Qiagen RNeasy Mini Kit (Qiagen) and the protocol was performed according to the manufacturer's instructions. For tissue RNA extraction, a homogenisation pre-step with stainless-steel beads and the Qiagen Tissuelyser LT (Qiagen) was carried out. Genomic DNA contamination was removed with the DNA-free removal kit (Ambion/Applied Biosystems, AM1906) and RNA concentration was quantified with the Nanodrop ND-100 spectrophotometer (Thermo Fisher).

**cDNA synthesis and quantitative qRT-PCR**. Here, 1 μg of RNA was reverse transcribed to cDNA using 4 μl of qScript cDNA SuperMix reagent (Quanta Bioscience, 95048–100) in a final reaction volume of 20 μl. PCR cycling conditions were as follows: 5 min incubation at 25 °C, DNA polymerisation at 42 °C for 30 min and enzyme deactivation at 85 °C for 5 min. PCR product was diluted 1:10. For the qRT-PCR experiment, a reaction mixture of 20 μl containing 10 μl of SYBR Master mix (Life Technologies, A25742), 0.5 μM of each reverse and forward primer, 5 μl of cDNA template and 4 μl of RNase-free water was prepared. The list of primers used are shown in Table S1. Reactions were conducted in duplicate and β-actin was used as a reference gene for CT-value normalisation. Amplification was conducted using the CFX Connect Real-Time System machine (Bio-Rad) and cycling conditions were as follows: a pre-incubation step of 95 °C at 15 min, followed by an amplification step of 95 °C for 10 s, 60 °C for 30 s and 72 °C for 30 s repeated for 44 cycles, and a melting curve analysis from 65 °C to 95 °C in 0.5 °C intervals.

**BioID protein interactome**. A plasmid encoding the BirA* enzyme tagged to a myc tag at the C-terminus was purchased from Addgene (Myc-BioID2-MCS plasmid, 74223) and purified by Miniprep (Qiagen) according to the manufacturer's protocol. Then, 5 μg of plasmid was digested with NheI-HF (NEB, R3131) and EcoRI-HF (NEB, R3101S) restriction enzymes to introduce the Myc-BioID2 sequence into the pLJM1-EGFP lentiviral vector (Addgene, 19319). Upon plasmid ligation and transformation, successful colonies were grown and purified by Miniprep. In parallel, *Rac1b* was purchased from GeneScript (OHu22224) and *Rac1* was amplified from normal murine small intestine tissue. Primers containing the sequences of Eco-RI-HF and SacII-HF (NEB, R0175S) restriction enzymes were used for *Rac1b* and *Rac1* amplification. Primer sequences were as follows: *Rac1/Rac1b* Forward (5′-3′): TTTTTGAATTCCAGGCCATCAAGTGTGTG and *Rac1/Rac1b* Reverse (5′-3′): TATATCCGCGGTTACAACAGCAGGCATTTTCTC. For amplification, Phusion DNA polymerase (NEB, M0530S) was used and the PCR protocol consisted on a pre-incubation step of 98 °C for 4 min, 27 amplification cycles composed of 45 s at 95 °C, 45 s at 59 °C, and 40 s at 72 °C and a final elongation step of 7 min at 72 °C. After an A-tailing reaction with the FastTaq polymerase (Roche, 12161508103) and 1 mM of dATP (PCR: 1 min at 95 °C and 30 min at 70 °C), the pGEM-T Easy Vector System (Promega, A1360) was utilised to insert *Rac1* and *Rac1b* sequences into the Myc-BioID2-containing lentiviral vector. Blue-white screening with X-gal/IPTG plates was used to detected positive (white) pGEM-T transformed colonies, which were grown and purified. PGEM-T insert was digested and sequenced for verification. Lentiviral vector was digested with EcoRI-HF and SacII-HF too and ligated with the pGEM-T inserts. Successful colonies were sequenced and amplified by HiSpeed Plasmid Maxi kit (Qiagen). As a BioID interaction control, an insert-free vector was also amplified. For the BioID pull-down experiment, a total of $3 \times 10^6$ cells of the CMT93 cell line were seeded in 10 cm$^2$ dish the day before transfection. Three plates were used per each condition (Rac1-BirA, Rac1b-BirA and BirA), 24 μg of DNA per plate was used for transfection, using 60 μg of Lipofectamine 2000 (Invitrogen, 11668030) and 40 h later, transfection medium was replaced with 50 μM of Biotin (1 mM biotin stock: 0.0122 g biotin dissolved in 50 ml DMEM) in 10 ml of cell medium (Sigma, B4501). Cells were incubated for 20 h and collected for protein extraction with 500 μl of RIPA buffer per dish. Upon protein clarification, the lysate was incubated with 15 μl of neutralised Streptavidin Sepharose matrix (GE Healthcare, 17–5113–01) for 6 h at 4 °C on a rotator. Following incubation, the lysate was spun down at 1000*g* for 5 min and supernatant was carefully removed. Beads were gently washed with 1 ml of wash buffer (50 mM TrisCl and 8 M Urea, pH 7.4) for 8 min at RT on a rotator and centrifuged at 1000*g* for 2 min. This washing step was repeated three times, the last wash with a urea-free buffer. Beads' pellets were a digested with Trypsin and processed as previously described[63]. Mass spectrometry was done using a Lumos Fusion (Thermo) mass spectrometer coupled to a RLS-nano uHPLC (Thermo). Peptides were separated by a 40 min linear gradient from 5% to 30% acetonitrile, 0.05% acetic acid. Proteins were identified and quantified the MaxQuant software suite using label-free quantification and searching against the mouse Uniprot database. Proteins enriched in Rac1-BirA, Rac1b-BirA vs BirA Protein were determined by >2-fold enrichment and *P* value < 0.05. NetworkAnalyst platform with the IMEx database was used for the interactome analysis of the resultant protein hits (networkanalyst.ca). To validate EGFR as a RAC1B-interacting protein, washed beads' pellets were resuspended with RIPA and 10X DTT, run on a western blot gel and probed against Streptavidin-HRP and EGFR. Lysate input was used as a positive control.

**Intestinal epithelial isolation and fractionation**. Harvested small intestine was washed with cold PBS, opened longitudinally and scrapped off with a coverslip to discard villi. The intestine was cut into small pieces and transferred to a 50 ml Falcon tube. Pieces were washed up and down several times with cold PBS, discarding supernatant with debris, and these were incubated with 25 ml of 2 mM EDTA for 30 min at 4 °C with agitation to allow crypt separation from surrounding tissue. EDTA was discarded and intestinal pieces were gently washed with cold PBS. Intestinal fractionation was initiated by adding 10 ml of cold PBS and thoroughly pipetting up and down several times. These 10 ml cell solutions were collected, which contained loosened intestinal cells, and corresponded to fraction number 1. This step was repeated three more times, the latter fractions being enriched in crypt/stem cells. For fractionation and transcriptomic analysis, fractions were independently spun down and pellets were used for RNA extraction and qRT-PCR experiments. However, when evaluation of overall epithelial intestinal cells was carried out, the four fractions were jointly collected and the cellular pellet was used for either RNA or protein extraction.

**Cells and cell culture**. The cell line used in this study was the mouse rectal carcinoma cell line CMT93[64]. A stock of $3–6 \times 10^6$ cells/ml was maintained over the study in T25 or T75 cell flasks (Corning) in Dulbecco's Modified Eagle's Medium (DMEM, Sigma) supplemented with 10% foetal calf serum and 1% of penicillin and streptomycin (Life Technologies, 15140). When cells reached 80% confluency, standard cell culture techniques were used for cell passaging and maintenance.

**EGF stimulation experiment**. In a 6-well plate, $2.5 \times 10^5$ CMT93 cells per well were seeded. The day after, a medium solution with 20 ng/ml EGF was prepared

and was added to the cells following 4 h serum starvation. Cells were collected after 5, 10, 15 and 30 min of EGF stimulation for protein extraction. A non-starved well and a non-EGF-stimulated well were used as controls in each experimental group.

**EGF uptake assay**. CMT93 cells were subjected to knockdown of *Rac1b* using mouse-specific PMOs targeting *Rac1b*, and a non-targeting control for 96 h. They were serum-starved for 6 h before being treated with EGF-AlexFluor555 (20 ng/ml) for the indicated times. Cells were then fixed in 4% PFA at room temperature (RT) for 15 min, before being quenched with 100 mM Glycine-PBS for 10 min. Following three washes with PBS, cells were permeabilised (0.2% saponin, 5% BSA, PBS pH 7.5) for 20 min at RT. Following 1 wash with IF buffer (0.2% Triton-X100, 0.05% Tween-20, PBS pH 7.5). Cells were then blocked in 3% BSA in IF buffer for 30 min, before being stained with α-EGFR (ab52894) in 1% BSA in IF buffer overnight at 4 °C. Following three washes in IF buffer, cells were stained with α-rabbit AlexaFluor488™ in 1% blocking solution for 45 min at RT. Cells were washed three times in IF buffer, before stained in Fluoromount-G® with DAPI. Images were taken on Nikon A1R confocal microscope. Images were analysed using ImageJ.

**EGFR-Rab5 co-localisation assay**. CMT93 cells stably expressing pBabe-mCherry-Rab5 construct (a gift from Dr. Noor Gammoh) were generated by retroviral transduction in DMEM high-glucose media containing (8 µg/ml polybrene). Two days post transduction, cells were selected for with puromycin (2 µg/ml) for 7 days. CMT93 cells were subjected to knockdown of *Rac1b* using mouse-specific PMOs targeting *Rac1b*, and a non-targeting control for 96 h. Cells were serum-starved for 6 h before being treated with EGF (20 ng/ml) for the indicated times. Cells were then fixed in 4% PFA at RT for 15 min, before being quenched with 100 mM Glycine-PBS for 10 min. Following three washes with PBS, cells were permeabilised (0.5% Triton X-100, PBS pH 7.5) for 20 min at RT. Following 1 wash with IF buffer (0.2% Triton-X100, 0.05% Tween-20, PBS pH 7.5). Cells were then blocked in 3% BSA in IF buffer for 30 min, before being stained with α-EGFR (ab52894) in 1% BSA in IF buffer overnight at 4 °C. Following three washes in IF buffer, cells were stained with α-rabbit AlexaFluor488™ in 1% blocking solution for 45 min at RT. Cells were washed three times in IF buffer, before being stained in Fluoromount-G® with DAPI. Images were taken on Nikon A1R confocal microscope. Co-localisation was measured using Coloc2 plugin on ImageJ.

**EGF tracking and lysosome-stained live cell imaging**. CMT93 cells were subjected to knockdown of *Rac1b* using mouse-specific PMOs targeting *Rac1b*, and a non-targeting control for 96 h. They were serum-starved for 6 h before treatment. Cells were stained with 25 nM LysoTracker™ Green DND-26 (Thermo Fisher Scientific) for 30 min at 37 °C, followed by two washes in Live Cell Imaging Solution (Invitrogen). Cell were then treated with 20 ng/ml EGF-AlexFluor555 and imaged every 30 s for the indicated times using an Andor spinning disk confocal microscope. EGF tracking was performed using TrackMate plugin on ImageJ. Co-localisation was performed by manual scoring of EGF-555/LysoTracker double-positive foci at each indicated time point. The total number of EGF-555 foci was determined using ImageJ "Find Maxima" at setting 100. Percentage EGF-555 foci co-localised with LysoTracker was calculated using these two values.

**Modulation of EGFR signalling in tumour organoids**. To inhibit EGFR with the PD153035 small inhibitor (Sigma, SML0564), organoids were washed and split as fragments by mechanically breaking the organoids with a P1000 instead of a P200. A density of 100 fragments/µl of BME was plated in 15 µl BME drops. Following 24 h of plating, growth media was replaced with media containing either 1 µM of PD15303 or 1% DMSO (Sigma) as control. Brightfield organoid pictures were taken following 20 h of treatment. To assess RAC1B dependency for EGFR signalling activation, organoids plated as fragments were grown either with EGF-free medium or with medium supplemented with EGF. Pictures were taken 72 h post plating. For both experimental setups, images were used to assess the organoid area using ImageJ.

**Cetuximab treatment of patient-derived organoids**. Human colorectal organoids derived from liver metastases (C001 and C002) were cultured in ADF medium containing 100 U/ml penicillin, 100 µg/ml streptomycin, 2 mM L-Glutamine, 10 mM HEPES, 1X Primocin, 1X B27, 1.25 mM NAC, 10 mM Nicotinamide, 50 ng/ml EGF, 500 nM A83-01, 10 µM SB202190, 100 ng/ml Noggin, 1 µM PGE and 10 nM gastrin. Organoids were passaged by breaking into small fragments and replating. *RAC1B* was depleted by treatment with 1 µM *hRAC1B* PMO or NT PMO for 96 h. After 96 h, the media was replaced with fresh media containing 1 µM *hRAC1B* PMO or NT PMO plus or minus 30 µg/ml cetuximab (Selleck, A2000). Pictures were taken 96 h post treatment. For all four experimental setups, images were used to assess the organoid area using ImageJ.

**Software**. The following software was used for data analysis: Microsoft Office Excel 365, GraphPad Prism v7.0 and v8.3.1, RNAseq analysis - TrimGalore 0.6.5, cutadapt 3.2, tophat 2.1.1 and cuffdiff 2.2.1, ImageJ 1.52p, kallisto 0.45, and DESeq2 v1.30.1.

**Reporting summary**. Further information on research design is available in the Nature Research Reporting Summary linked to this article.

## Data availability statement

RNAseq data that support the findings in this manuscript have been deposited at Gene Expression Omnibus (NCMI) with the study accession code: GSE167876. Source data are provided with this paper.

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

## Acknowledgements

We thank the core technical services at the Institute of Genetics and Molecular Medicine. In particular the staff of the Bioresearch & Veterinary Services and Dr. Ann Wheeler and Dr. Laura Murphy from the Advanced Imaging Resource (AIR) for their assistance. We thank the patient / recruiting nursing team at the Western General Hospital for providing the tumour material used in this study. Patient material was obtained under ethical consent SOCCS3 (REC: 11/SS/0109). K.B.M, V.G., C.B., P.C. and A.E.H. are supported by a CRUK Career Development Fellowship A19166 and an ERC Starting Grant 715782. O.J.S. is supported by CRUK grants A26825, A28233, A23390, A21139, A12481 and A17196 and an ERC Starting Grant 311301. F.V.N.D. was supported by Clinical Scientist Fellowships from CRUK (C26031/A11378) and Chief Scientist Office (SCAF/16/01). A.B and A.v.K carried out Mass Spectrometry analysis supported by the Wellcome Trust (Multiuser Equipment Grant, 208402/Z/17/Z). Fig. S8 was created with BioRender.

## Author contributions

Study concept and design: K.B.M. Acquisition of data: V.G., C.V.B., P.C., A.B., S.O.P., C. N., A.E.H., A.W., L.M. and K.B.M. Analysis and interpretation of data: V.G., C.V.B., P.C., A.B., S.A., S.O.P., M.A., J.C., A.v.K., A.W., L.M. and K.B.M. Drafting of manuscript: K.B. M., V.G., S.O.P., P.C. and A.E.H. Material support: D. Ste, D. Str, P.F., M.D., F.D., L.P., G. D. and O.J.S. Design of PMO: L.P. and G.D.

## Competing interests

The authors declare no competing interests.
