## [Peer Review File · Nature Communications]

Reviewers' Comments:

Reviewer #1:

Remarks to the Author:

The authors had already demonstrated that RAC1 signaling is critical for the expansion of intestinal stem cells and subsequent tumor formation following Apc deletion in the mouse (Myant KB et al cell Stem Cell.2013). Now they find that the RAC1 variant RAC1B (containing an additional exon 4) is over-represented in high-grade tumors and correlates with poor prognosis inside patients with comparable tumor stage (and subtype). Mechanistically, they identify a link between RAC1B and EGFR signaling and propose the possibility of targeting RAC1B to increase the therapeutic potency of EGFR inhibitors in cancer. The concept is interesting and clinically relevant. However, there are several issues that should be revised to re-enforce authors' conclusions.

1-In situ hybridization of RAC1B isoform shown in figure 2 is difficult to visualize. Maybe showing higher magnification images without circles could help in the interpretation. Moreover, it is not clear to me why only few cells in the wild-type contain RAC1B (excluding the putative Lgr5+ stem cells localized at the bottom of the crypts) and why it is detected in the villus of APCfl/fl. Is this because APCfl/fl villus display high levels of nuclear beta-catenin that impose RAC1B expression? It would be helpful to show double detection of nuclear beta-catenin (IHC) and RAC1B (ISH) but also detection of RAC1B in the Lgr5+ reporter mice to see whether normal or cancer ISCs express RAC1B.

2- Related with figure 3, authors mention that "half of the tumors in the APCfl;RAC1B deleted mice expressed high levels of Rac1b (Fig S3B - C). This indicated a failure to efficiently delete Rac1b suggesting a positive selective pressure for maintaining Rac1b expression in tumor initiating cells". But then, they say: "However, tumors from Apc Rac1b mice were significantly less proliferative than controls". To my view, this means that tumors in the compound mice strain proliferate less independently of the extent of RAC1B deletion. Authors should clarify this by showing ki67 or BrdU incorporation together with RAC1B detection (in double staining or in consecutive sections) to confirm the direct association between RAC1B and tumor cell proliferation.

3-In figure 5 the EGFR pathway is identified as a common interactor of RAC1 and RAC1B but then authors suggest and show data indicating a preferential interaction between RAC1B and EGF, which is a little contradictory. In fact, the results from BioID experiments suggest that RAC1 and RAC1B interactors are primarily identical. Why authors do not focus and confirm the differential binding of the 8 proteins that seem to specifically interact with RAC1B? If confirm, the functional relevance of this specific RAC1B interactors should be tested. Then, authors investigate the possibility that RAC1B depletion affects EGFR activation by EGF in vitro and show a moderate decrease in signaling. However, a mechanistic explanation on how RAC1B controls EGFR signaling is lacking. Do the authors have any idea on how this is working? If, as mentioned in the discussion section, RAC1B controls EGFR recycling, this needs to be demonstrated.

4-In the final figures, it is suggested that targeting RAC1B together with EGFR inhibitors reduces tumor growth, which could be clinically relevant. A more clinically relevant model needs to be included in the paper to make this possibility more appealing.

Reviewer #2:

Remarks to the Author:

In this manuscript, Dr. Myant and colleagues proposed RAC1B as a potential therapeutic target for enhancing EGFR inhibitor efficacy in CRC. They utilized patient data, in vivo mouse models and organoid culture system to study the functional importance of RAC1B in CRC. Overall, this study was well designed and provides interesting findings about CRC. However, there are several concerns to be addressed before being considered for publication in Nat Communications.

1. From my personal perspective, "EGFR inhibitor resistant colorectal cancer" in the manuscript

title is somehow misleading. Throughout the study, the CRC models that the authors used are mainly loss-of-APC or Wnt activation driven mouse models or organoids. Patients with CMS2 (Wnt activated) tumors preferentially benefit from anti-EGFR therapy. Sensitizing treatment-naive tumor cells to EGFR inhibitor is different from targeting EGFR inhibitor resistant cancer. With the current title, I would expect to see more data from tumors with acquired resistance to anti-EGFR/EGFR inhibitor therapy in CRC, as compared to the treatment-naive tumors.

2. In Figure 1E, it would be more helpful to show the RAC1B PSI individually in all four subtypes.

3. In Figure 2D, n=3 mice for quantification is a relatively small number. Also how many areas in each animal were used for probe positivity counting? This detail should be included in figure legend.

4. In the Apc Rac1b mouse model, the authors reported that around half of the tumors expressed high levels of Rac1b (Fig S3B-C). Therefore in Figure 3F-G and Figure S3E-F, is the Apc Rac1b tumor sample with efficient Rac1b deletion or still with high Rac1b expression? This is very important and the Rac1b expression data should be shown clearly. Related to this, is there difference in tumor cell proliferation between Apc Rac1b tumors with high Rac1b expression and Apc Rac1b tumors with efficient Rac1b deletion? Similarly, in Figure 3H and Figure S4, was the Rac1b efficiently deleted?

5. Referring to the above point, the authors did a good job with the RNA-seq experiment because they confirmed negative for Rac1b by qRT-PCR in Apc Rac1b tumor samples used for RNA-seq study. The finding that Rac1b is required for oncogenic Wnt signaling is striking. However, the mechanism was overall less well developed throughout the study. How does Rac1b regulate oncogenic Wnt signaling downstream of beta-catenin nuclear localization? The authors talked about the link between EGFR and WNT9A-FZD9B-LRP complex, but this could not provide any insight into the Rac1b mechanism given that the authors used Apc-loss model which does not require the membrane Wnt receptor-coreceptor to activate Wnt signaling.

6. In Figure 6E and 6F, it would be nice to show western blot analyses of EGFR, AKT, ERK and their phosphorylation in all four treatment conditions.

7. To support the discussed EGFR-RAC1B feed forward loop, it would be nice to show in the organoid model, whether inhibiting EGFR signaling could reduce the Rac1b expression and consequently decrease Wnt signaling.

Reviewer #3:

Remarks to the Author:

The manuscript by Gudino V et al investigates the role of RAC1b in colorectal cancer. The current manuscript builds on previous work from the Myant and Sansom laboratories published in Cell Stem Cell (2013). In their earlier study, the group reported that RAC1 was an important downstream target of Wnt signaling, and helped to promote intestinal stem cell expansion and intestinal tumorigenesis through ROS production and NF- κ B activation. Here, the authors focus on RAC1b, a splice variant generated by inclusion of exon 4 that has constitutive activity and is overexpressed in numerous tumor types. They show that RAC1b overexpression correlates with Wnt activity and poor prognosis, is expressed in intestinal crypts and increased following Apc loss, and that deletion of Rac1b in Apc mutant mice inhibits tumorigenesis. They go on to use a number of in vitro cell lines and organoid models to show that RAC1B likely interacts with EGFR signaling components and thus that knockdown of RAC1B using morpholinos can increase responses to EGFR inhibitors in vitro. Overall, this study extends our understanding of RAC1 and contains some interesting data and observations. However, the study also contains a number of inconsistencies and weaknesses, and falls short of convincing that RAC1b represents a viable therapeutic target. Major concerns are listed below.

1. Overexpression of RAC1B and correlation with poor prognosis.

While data is presented on overexpression of RAC1B in CRC, these increases are modest. In addition, while there is an overall increase in RAC1B high tumors late stage (III & IV) and with survival, the finding that RAC1B high tumors are more common in CMS2 tumors which have a relatively good prognosis is confusing and not well explained. In addition, the "cutoffs" for RAC1B

high tumors seems to result in a majority of tumor samples showing high expression. In fact, the RAC1B high tumors are more common in all stages (S1C). Is it the case that only a small minority of tumors are RAC1B low? While the authors show a correlation of RAC1B with gene expression in specific subtypes, what is not shown is the percentage (%) of RAC1B high tumors in each of the tumor subsets. It does seem that RAC1B overexpression is more common in the tumors with the best prognosis. Overall, this presentation is quite confusing and it is difficult to understand this correlation with prognosis.

2. RAC1B expression localization and correlation with Wnt activity.

The data in Fig. 2 is focused on the localization of RAC1B expression, which suggests that RAC1B is expressed primarily in the crypts, particularly in the lowest crypt cells such as fraction 1B, which seems to correlate with Lgr5 and other stem cell markers. Indeed, in the previous publication (Myant et al, 2013) their conclusion was that RAC1 was downstream of Wnt signaling and required for ISC activity. However, in the RNAscope studies, the images shown suggest a less specific pattern of localization, with expression above the base of the crypts. In addition, there appears to be no correlation in RAC1 expression with Lgr5+ cells here, in contrast to the conclusion of the earlier publication. It would seem to be important to provide data that RAC1B is more highly expressed in Lgr5+ cells in order to promote a cell autonomous mechanism for RAC1B. The investigators could in theory sort Lgr5-EGFP and compare Rac1b expression in Lgr5+ and Lgr5- cells. Alternatively they could show by RNAscope Rac1b expression in Lgr5+ cells. Does Rac1b influence Lgr5 cell tracing as was the case for RAC1, or is it only the non-RAC1B isoforms that regulate ISCs. Furthermore, the notion that RAC1B expression is a downstream target of Wnt is not addressed in a cell specific manner here. Ideally, the investigators could show in Apc deficient mice expression of Rac1b in cells that are positive for nuclear beta-catenin. The data in its present form do show correlations between Wnt activity and RAC1B and crypt progenitor cell proliferation, but not necessarily in a direct and cell autonomous manner.

3. Rac1b deletion in Apc mutant mice.

Homozygous knockout of Rac1b does appear to lead to slower tumor growth and prolonged the survival of Vil-Apc f/+ heterozygous mice. These results are somewhat similar to earlier studies with Rac1 conditional deletion, although the differences compared to Rac1 KO are not addressed or directly compared. However, it is curious that the studies here were carried out in the Apc heterozygous KO mice rather than the Apc homozygous F/F animals. Was there not a difference in survival with Rac1b deletion in the homozygous Vil-Apc f/f animals, since these homozygous were clearly generated (or the later organoid studies). Second, the analysis seems largely confined to the small intestine; was there any effect on colonic adenomas in this model?

4. Effect of RAC1B on EGFR signaling.

BioID experiments suggest that RAC1B interacts with EGFR signaling components and biochemical studies and antisense studies in cell lines and organoids are consistent with this sort of pathway. However, no evidence is provided from the animal models to support this. At a minimum, one would like to see staining for phosphor-EGFR and phospho-ERK to demonstrate that deletion of RAC1B modulates EGFR signaling in vivo.

5. PDO studies.

The investigators show that knockdown of RAC1B reduces clonogenic growth of PDOs from benign polyps but not from invasive tumors, although the latter could be suppressed by the addition of an EGFR inhibitor. However, the concern here is that this appears to be a single set of tumors, with limited studies and endpoints, and it is not clear if this is a RAC1B high or low tumor. While the morpholino approach is interesting, and seems to be the "therapeutic strategy" highlighted in the title, there is a conspicuous absence of in vivo preclinical therapeutic models here. Evidence that RAC1B knockdown in combination with EGFR inhibition in vivo is needed to support the title of the study. In addition, the title is a bit confusing; was RAC1B knockdown not effective in EGFR inhibitor "sensitive" colorectal cancers?

6. Wnt signaling.

Finally, it is somewhat confusing that RAC1B is both downstream and upstream of Wnt signaling and this does not seem to be sufficiently clarified in the manuscript. How is RAC1B activated by Wnt signaling? And how does it then further enhance the Wnt pathway? The previous publication by Myant et al suggested that ROS and NF-kB were important pathways downstream of RAC1 but

these are not mentioned here. Are they also relevant to RAC1B? The model figure (Fig. 7F) seems to suggest that RAC1B has two independent actions: (i) enhancing Wnt activity and (ii) enhancing EGFR and ERK activation. Given that previous studies have indicated that EGFR activation can enhance Wnt activation; is it possible that EGFR activation is responsible for the enhanced Wnt activation? If so, then the model figure needs to be revised.

We thank the three reviewers for taking the time to review our manuscript and for their insightful analysis and comments. We have outlined our responses below each individual comment in bold. In addition, changes to the manuscript text are referred to in our responses and are highlighted in yellow in the revised manuscript.

Reviewer #1 (Remarks to the Author):

The authors had already demonstrated that RAC1 signaling is critical for the expansion of intestinal stem cells and subsequent tumor formation following *Apc* deletion in the mouse (Myant KB et al *Cell Stem Cell*.2013). Now they find that the RAC1 variant RAC1B (containing an additional exon 4) is over-represented in high-grade tumors and correlates with poor prognosis inside patients with comparable tumor stage (and subtype). Mechanistically, they identify a link between RAC1B and EGFR signaling and propose the possibility of targeting RAC1B to increase the therapeutic potency of EGFR inhibitors in cancer. The concept is interesting and clinically relevant. However, there are several issues that should be revised to re-enforce authors' conclusions.

We thank the reviewer for their supportive comments and outline our responses in detail below.

1-In situ hybridization of RAC1B isoform shown in figure 2 is difficult to visualize. Maybe showing higher magnification images without circles could help in the interpretation. Moreover, it is not clear to me why only few cells in the wild-type contain RAC1B (excluding the putative *Lgr5+* stem cells localized at the bottom of the crypts) and why it is detected in the villus of *APCfl/fl*. Is this because *APCfl/fl* villus display high levels of nuclear beta-catenin that impose RAC1B expression? It would be helpful to show double detection of nuclear beta-catenin (IHC) and RAC1B (ISH) but also detection of RAC1B in the *Lgr5+* reporter mice to see whether normal or cancer ISCs express RAC1B.

We have included new, higher magnification images for Figure 2D to help interpretation of the data. The reviewer is correct that the expression of *Rac1b* is low in normal tissue and thus detection by the Basescope probe is limited to a small number of cells. In order to investigate the link between *Rac1b* expression and *Lgr5+* stem cells in more detail we have carried out qRT-PCR analysis of *Rac1b* in sorted *Lgr5+* and *Lgr5-* cells and scored the crypt position of *Rac1b* expressing cells. This data is included as Figure 2C and Figure S2B+E. This new data suggests that *Rac1b* expression is enriched in *Lgr5+* stem cells and *Rac1b* expressing cells are found primarily in crypt cell positions. However, the enrichment is not as strong as *Lgr5* expression or completely restricted to the crypt base positions (Figure S2B+D) leading us to conclude that *Rac1b* is enriched in proliferative crypt cells but is not a well defined markers of *Lgr5+* stem cells. The reviewer is also correct that the reason *Rac1b* is detected in the villus of *Apc* deficient mice is due to high levels of nuclear beta-catenin. Unfortunately, we were technically unable to carry out dual nuclear beta-catenin / *Rac1b* detection due to the protocols being incompatible but have included images of beta-catenin IHC from the WT and *Apc fl/fl* mice used for Basescope analysis. These clearly show that following *Apc* deletion all cells in the villus contain robust nuclear accumulation of beta catenin and this correlates with detection of *Rac1b* by Basescope in the villus (Figure S2E and Figure 2D-F).

2- Related with figure 3, authors mention that “half of the tumors in the APC^{fl};RAC1B deleted mice expressed high levels of Rac1b (Fig S3B - C). This indicated a failure to efficiently delete Rac1b suggesting a positive selective pressure for maintaining Rac1b expression in tumor initiating cells”. But then, they say: “However, tumors from Apc Rac1b mice were significantly less proliferative than controls”. To my view, this means that tumors in the compound mice strain proliferate less independently of the extent of RAC1B deletion. Authors should clarify this by showing ki67 or BrdU incorporation together with RAC1B detection (in double staining or in consecutive sections) to confirm the direct association between RAC1B and tumor cell proliferation.

We thank the reviewer for this insightful comment and agree it is an important question to address. Using *Rac1b* Basoscope we analysed our *Apc Rac1b* mice for tumours that had not efficiently deleted *Rac1b* (Figure S3E). By staining these tumour tumours for BrdU incorporation we found that these tumours proliferate significantly better than those that are *Rac1b* negative and to a similar extent as tumours from *Apc* control mice. Thus, there is a direct association between RAC1B and tumour cell proliferation. These experiments are described in the manuscript on page 6.

3-In figure 5 the EGFR pathway is identified as a common interactor of RAC1 and RAC1B but then authors suggest and show data indicating a preferential interaction between RAC1B and EGF, which is a little contradictory. In fact, the results from BioID experiments suggest that RAC1 and RAC1B interactors are primarily identical. Why authors do not focus and confirm the differential binding of the 8 proteins that seem to specifically interact with RAC1B? If confirm, the functional relevance of this specific RAC1B interactors should be tested. Then, authors investigate the possibility that RAC1B depletion affects EGFR activation by EGF in vitro and show a moderate decrease in signaling. However, a mechanistic explanation on how RAC1B controls EGFR signaling is lacking. Do the authors have any idea on how this is working? If, as mentioned in the discussion section, RAC1B controls EGFR recycling, this needs to be demonstrated.

The reviewer makes an interesting point regarding the similarity of the Rac1 and Rac1b interactomes. Although one option would have been to investigate proteins that interact specifically with Rac1b we felt that the interaction with multiple components of the EGFR signalling network warranted further investigation. In particular, as the EGFR signalling pathway is a therapeutic target in colorectal cancer, our findings would potentially be relevant to the treatment of this disease. In addition, as the interactomes of Rac1 and Rac1b were so similar it suggests that the two isoforms are likely to have similar functions with the key difference between them being the constitutive, RacGEF independent activity of Rac1b. Thus, we felt there was a strong rationale for investigating this further. We have included this strengthened argument to support the investigation of EGFR signalling on page 7 of the manuscript. To provide mechanistic insight into how RAC1B controls EGFR signalling we investigated EGFR trafficking using fluorescent labelled EGF and live cell imaging. We found that EGF stimulated EGFR trafficking was reduced upon RAC1B depletion (Figure 6A-C and S6C). Whilst RAC1B depletion did not alter colocalisation of EGFR with early endosomes (Figure S6C) we observed increased accumulation of EGFR in lysosomes (Figure 6D-E and S6D). Together, this suggests RAC1B is required for efficient trafficking and recycling of EGFR. These data are outlined in the highlighted section on page 8.

4-In the final figures, it is suggested that targeting RAC1B together with EGFR inhibitors reduces tumor growth, which could be clinically relevant. A more clinically relevant model needs to be included in the paper to make this possibility more appealing.

The reviewer makes an excellent point and to address this we obtained cetuximab resistant human colorectal cancer liver metastatic organoids from the Valeri lab (Vlachogiannis et al., 2018, PMID: 29472484). Two organoid lines were tested, C001 and C002, both are Kras WT and both are resistant to cetuximab treatment *in vitro*. In addition C002 was derived from a patient who acquired resistance to cetuximab upon treatment. Treatment of both organoid lines with our *RAC1B* inhibitor sensitised them to cetuximab treatment (Figure 7E-F and S7G-H). Thus, using a clinically relevant model we demonstrate the potential for *RAC1B* inhibition to enhance the efficacy of a clinically utilised EGFR inhibitor. These data are outlined in the highlighted section on page 9.

Reviewer #2 (Remarks to the Author):

In this manuscript, Dr. Myant and colleagues proposed RAC1B as a potential therapeutic target for enhancing EGFR inhibitor efficacy in CRC. They utilized patient data, *in vivo* mouse models and organoid culture system to study the functional importance of RAC1B in CRC. Overall, this study was well designed and provides interesting findings about CRC. However, there are several concerns to be addressed before being considered for publication in Nat Communications.

We thank the reviewer for their supportive comments and provide details of our response to the points below.

1. From my personal perspective, “EGFR inhibitor resistant colorectal cancer” in the manuscript title is somehow misleading. Throughout the study, the CRC models that the authors used are mainly loss-of-APC or Wnt activation driven mouse models or organoids. Patients with CMS2 (Wnt activated) tumors preferentially benefit from anti-EGFR therapy. Sensitizing treatment-naive tumor cells to EGFR inhibitor is different from targeting EGFR inhibitor resistant cancer. With the current title, I would expect to see more data from tumors with acquired resistance to anti-EGFR/EGFR inhibitor therapy in CRC, as compared to the treatment-naive tumors.

The reviewer makes an excellent point and we agree that additional evidence was required to support the manuscript title. To address this we obtained cetuximab resistant human colorectal cancer liver metastatic organoids from the Valeri lab (Vlachogiannis et al., 2018, PMID: 29472484). Two organoid lines were tested, C001 and C002, both are Kras WT and both are resistant to cetuximab treatment *in vitro*. In addition, C002 was derived from a patient who acquired resistance to cetuximab upon treatment. Treatment of both organoid lines with our *RAC1B* inhibitor sensitised them to cetuximab treatment (Figure 7E-F and S7G-H). Thus, using a clinically relevant model we demonstrate the potential for *RAC1B* inhibition to enhance the efficacy of a clinically utilised EGFR inhibitor providing additional evidence to support the conclusions of our manuscript. These data are outlined in the highlighted section on page 9.

2. In Figure 1E, it would be more helpful to show the RAC1B PSI individually in all four subtypes.

We have included this data as Figure S1F.

3. In Figure 2D, n=3 mice for quantification is a relatively small number. Also how many areas in each animal were used for probe positivity counting? This detail should be included in figure legend.

The reviewer is correct that n=3 is the minimum number of mice required for this experiment. However, the data showing increased *Rac1b* expression following *Apc* deletion in this experiment are highly significant ($p < 0.0001$). In addition, these data are supported by qRT-PCR analysis from several different mouse models (Figure S2F and S3C) and analysis of human TCGA expression data (Figure 1A). We have included details of how the scoring was carried out in the figure legend. Briefly, for each mouse, at least 5 areas for crypt and villus regions were scored. In total these areas incorporated at least 50 crypt/villus axes and covered $\sim 2\text{mm}^2$ total area.

4. In the *Apc Rac1b* mouse model, the authors reported that around half of the tumors expressed high levels of *Rac1b* (Fig S3B-C). Therefore in Figure 3F-G and Figure S3E-F, is the *Apc Rac1b* tumor sample with efficient *Rac1b* deletion or still with high *Rac1b* expression? This is very important and the *Rac1b* expression data should be shown clearly. Related to this, is there difference in tumor cell proliferation between *Apc Rac1b* tumors with high *Rac1b* expression and *Apc Rac1b* tumors with efficient *Rac1b* deletion? Similarly, in Figure 3H and Figure S4, was the *Rac1b* efficiently deleted?

We thank the reviewer for this insightful comment and agree that determining which tumours still express *Rac1b* in our *Apc Rac1b* mice is an important question to address. To clarify our analysis we used *Rac1b* Basoscope to identify tumours from our *Apc Rac1b* mice that had not efficiently deleted *Rac1b* (Figure S3E). By staining these tumours for BrdU incorporation we found that these tumours proliferate significantly better than those from the same mice that are *Rac1b* negative. In fact the proliferative rate of these tumours is similar to tumours from *Apc* control mice. Thus, there is a direct association between *RAC1B* and tumour cell proliferation. These experiments are described in the manuscript on page 6. In addition, all analyses carried out on tumours from these mice (Figures S3F, G and S4A) were carried out on tumours confirmed to be *Rac1b* deleted. This is now referred to in the figure legends. Also, using qRT-PCR we confirmed that the organoids used in Figure 3H were *Rac1b* deleted (Figure S3K-L).

5. Referring to the above point, the authors did a good job with the RNA-seq experiment because they confirmed negative for *Rac1b* by qRT-PCR in *Apc Rac1b* tumor samples used for RNA-seq study. The finding that *Rac1b* is required for oncogenic Wnt signaling is striking. However, the mechanism was overall less well developed throughout the study. How does *Rac1b* regulate oncogenic Wnt signaling downstream of beta-catenin nuclear localization? The authors talked about the link between EGFR and WNT9A-FZD9B-LRP complex, but this could not provide any insight into the *Rac1b* mechanism given that the authors used *Apc*-loss model which does not require the membrane Wnt receptor-coreceptor to activate Wnt signaling.

The reviewer makes an interesting point regarding the mechanism of *Rac1b* regulated Wnt signalling. We have investigated this in more detail, in particular the potential link between EGFR signalling and Wnt signalling in our model systems. Firstly, we investigated the impact of EGFR inhibition on Wnt target gene expression in *Apc* deficient organoids and found that, similar to following *Rac1b* deletion, a subset of targets were downregulated (Figure S6F). In addition, we depleted *Rac1b* in *Apc* deficient organoids carrying an additional activating mutation of *Kras*. In

these organoids, where MAPK signalling is constitutively active, downstream of EGFR, the depletion of *Rac1b* had no effect on organoid clonogenic capacity or Wnt target gene expression (Figure S6G-J). Together, these data suggest there is cross talk between Wnt and EGFR signalling pathways and *Rac1b* mediated control of EGFR signalling activation is required for efficient Wnt signalling activity. As this occurs in *Apc* deficient organoids it suggests this occurs downstream of beta-catenin nuclear localisation. Defining the precise mechanism of this regulation will be an interesting avenue for future investigation. These data are referred to on page 8 of the new manuscript.

6. In Figure 6E and 6F, it would be nice to show western blot analyses of EGFR, AKT, ERK and their phosphorylation in all four treatment conditions.

We were unable to extract high quality protein from the 24h timepoint used in the experiment as by this time the majority of *Apc Rac1b* organoids are dead. As an alternative, we carried out IHC for pEGFR and pERK (pAKT IHC showed no detectable signal and is not shown) on fixed organoids from this experiment at an earlier timepoint (6h). These data are included as Figure S7B.

7. To support the discussed EGFR-RAC1B feed forward loop, it would be nice to show in the organoid model, whether inhibiting EGFR signaling could reduce the *Rac1b* expression and consequently decrease Wnt signaling.

We investigated the potential EGFR-*Rac1b* feed forward loop however, EGFR inhibition did not lead to reduced *Rac1b* expression in *Apc* deficient organoids (Figure S6F). Therefore, we have removed this section from the discussion.

Reviewer #3 (Remarks to the Author):

The manuscript by Gudino V et al investigates the role of RAC1b in colorectal cancer. The current manuscript builds on previous work from the Myant and Sansom laboratories published in Cell Stem Cell (2013). In their earlier study, the group reported that RAC1 was an important downstream target of Wnt signaling, and helped to promote intestinal stem cell expansion and intestinal tumorigenesis through ROS production and NF-kB activation. Here, the authors focus on RAC1b, a splice variant generated by inclusion of exon 4 that has constitutive activity and is overexpressed in numerous tumor types. They show that RAC1b overexpression correlates with Wnt activity and poor prognosis, is expressed in intestinal crypts and increased following *Apc* loss, and that deletion of *Rac1b* in *Apc* mutant mice inhibits tumorigenesis. They go on to use a number of in vitro cell lines and organoid models to show that RAC1B likely interacts with EGFR signaling components and thus that knockdown of RAC1B using morpholinos can increase responses to EGFR inhibitors in vitro. Overall, this study extends our understanding of RAC1 and contains some interesting data and observations. However, the study also contains a number of inconsistencies and weaknesses, and falls short of convincing that RAC1b represents a viable therapeutic target. Major concerns are listed below.

We thank the reviewer for their insightful analysis. Specific responses to the reviewer's comments are outlined below.

1. Overexpression of RAC1B and correlation with poor prognosis.

While data is presented on overexpression of RAC1B in CRC, these increases are modest. In addition, while there is an overall increase in RAC1B high tumors late stage (III & IV) and with survival, the finding that RAC1B high tumors are more common in CMS2 tumors which have a relatively good prognosis is confusing and not well explained. In addition, the “cutoffs” for RAC1B high tumors seems to result in a majority of tumor samples showing high expression. In fact, the RAC1B high tumors are more common in all stages (S1C). Is it the case that only a small minority of tumors are RAC1B low? While the authors show a correlation of RAC1B with gene expression in specific subtypes, what is not shown is the percentage (%) of RAC1B high tumors in each of the tumor subsets. It does seem that RAC1B overexpression is more common in the tumors with the best prognosis. Overall, this presentation is quite confusing and it is difficult to understand this correlation with prognosis.

We apologise if the presentation of this data was confusing and have sought to clarify our analyses:

- 1) Regarding the finding that Rac1b is highly expressed in CMS2 tumours and correlates with poor prognosis. The reviewer is correct that this appears contradictory as overall CMS2 tumours have relatively good prognosis. However, a significant proportion of CMS2 tumours have poor prognosis and these are the ones that express high levels of Rac1b. Therefore, high Rac1b expression marks a subset of CMS2 tumours with poor prognosis. We have clarified this in the text on page 4 in the third highlighted section.**
- 2) The cutoff for Rac1b high tumours is shown in Figure S1B (red shaded part of the graph) and in fact only ~18% of tumours are Rac1b high so only a minority of tumours are Rac1b high. The reviewer is correct that a small minority of tumours are Rac1b low so we have repeated the analysis of Figure S1C-D comparing Rac1b high tumours to tumours in both the low and intermediate RAC1B expression groups. We find that Rac1b high tumours are significantly more prevalent in late stage disease and in tumours with increased lymphovascular invasion compared to the lower RAC1B expression groups (Figure S1C-D). We apologise for the lack of clarity in this section and hope that this is now clearer.**
- 3) We have included the percentage of Rac1b high tumours in each tumour subset (Figure S1G-H). Interestingly, Rac1b high tumours are also quite prevalent in CMS4, again supportive of a role in late stage disease.**
- 4) This section has been rewritten to improve clarity (page 4 highlighted sections).**

2. RAC1B expression localization and correlation with Wnt activity.

The data in Fig. 2 is focused on the localization of RAC1B expression, which suggests that RAC1B is expressed primarily in the crypts, particularly in the lowest crypt cells such as fraction 1B, which seems to correlate with Lgr5 and other stem cell markers. Indeed, in the previous publication (Myant et al, 2013) their conclusion was that RAC1 was downstream of Wnt signaling and required for ISC activity. However, in the RNAscope studies, the images shown suggest a less specific pattern of localization, with expression above the base of the crypts. In addition, there appears to be no correlation in RAC1 expression with Lgr5+ cells here, in contrast to the conclusion of the earlier publication. It would seem to be important to provide data that RAC1B is more highly expressed in Lgr5+ cells in order to promote a cell autonomous mechanism for RAC1B. The investigators could in theory sort Lgr5-EGFP and compare Rac1b expression in Lgr5+ and Lgr5- cells. Alternatively they could show by RNAscope Rac1b expression in Lgr5+ cells. Does Rac1b influence Lgr5 cell tracing as was the case for RAC1, or is it only the non-RAC1B isoforms that regulate ISCs. Furthermore, the notion that RAC1B expression is a downstream target of Wnt is not addressed in a cell specific manner here. Ideally, the investigators could show in Apc deficient mice expression of Rac1b in cells

that are positive for nuclear beta-catenin. The data in its present form do show correlations between Wnt activity and RAC1B and crypt progenitor cell proliferation, but not necessarily in a direct and cell autonomous manner.

The reviewer raises a number of important points, which we have addressed as follows:

- 1) In order to investigate the link between *Rac1b* expression and Lgr5+ stem cells in more detail we have carried out qRTPCR analysis of *Rac1b* in sorted Lgr5+ and Lgr5- cells and scored the crypt position of *Rac1b* expressing cells. This data is included as Figure 2C and Figure S2B+D. This new data suggests that *Rac1b* expression is enriched in Lgr5+ stem cells and *Rac1b* expressing cells are found primarily in crypt cell positions. However, the enrichment is not as strong as Lgr5 expression (Figure S2B) or completely restricted to the crypt base positions (Figure S2D) leading us to conclude that *Rac1b* is enriched in proliferative crypt cells but is not a well defined marker of Lgr5+ stem cells (outlined in the first 2 highlighted sections on page 5 of the manuscript).
- 2) We did not directly measure the impact of *Rac1b* deletion on Lgr5 lineage tracing. However, we believe that *Rac1b* deletion does not detrimental affect the function of normal Lgr5+ ISC cells as normal intestinal homeostasis is not perturbed by *Rac1b* deletion (Figure S3H-I) and *Rac1b* deletion is maintained in normal tissue for at least 6 months (Figure S3C, compare *WT* to *Rac1b*). In addition, it is worth noting that in our previous publication (Myant et al., 2013) on the function of Rac1, normal ISC function was not impaired upon *Rac1* deletion either. It was only upon *Apc* deletion that *Rac1* was required for transformed Lgr5+ ISCs to expand and proliferate out with their niche.
- 3) Unfortunately, we were technically unable to carry out dual nuclear beta-catenin / *Rac1b* detection to determine directly this association due to the staining protocols being incompatible. However, we have included images of beta-catenin IHC from the *WT* and *Apc* fl/fl mice used for the Basescope analysis. These clearly show that following *Apc* deletion all cells in the crypts and villus contain robust nuclear accumulation of beta catenin and this correlates with detection of *Rac1b* by Basescope in this tissue (Figure S2E and Figure 2D-F). Although the data are supportive of a direct and cell autonomous regulation the reviewer is correct that this may not necessarily be the case so we have clarified in the text that there is a correlation between *Rac1b* expression and activated Wnt signalling (page 5, 4th highlighted section).

3. *Rac1b* deletion in *Apc* mutant mice.

Homozygous knockout of *Rac1b* does appear to lead to slower tumor growth and prolonged the survival of *Vil-Apc* f/+ heterozygous mice. These results are somewhat similar to earlier studies with *Rac1* conditional deletion, although the differences compared to *Rac1* KO are not addressed or directly compared. However, it is curious that the studies here were carried out in the *Apc* heterozygous KO mice rather than the *Apc* homozygous F/F animals. Was there not a difference in survival with *Rac1b* deletion in the homozygous *Vil-Apc* f/f animals, since these homozygous were clearly generated (or the later organoid studies). Second, the analysis seems largely confined to the small intestine; was there any effect on colonic adenomas in this model?

We chose to analyse *Apc* heterozygous animals as these give a better representation of intestinal tumourigenesis than the short term acute deletion in *Apc* flox/flox animals. The reviewer is correct that *Apc* flox/flox mice were generated for the organoid study but they were sampled at a day 5 post induction timepoint and survival was not analysed. Analysis of tissue from these mice

showed that proliferation was modestly reduced in the *Rac1b* deleted mice but this was not statistically significant (reviewer figure below). A possible explanation for this is that *Rac1b* is more important for cellular transformation and tumour growth / expansion when a tumour initiates from a single *Apc* deficient cell (as would occur in the *Apc* heterozygous model). As this is a more accurate representation of human tumourigenesis we feel this is a better model to use for this study.

A

Reviewer Figure. Proliferation analysis of *Apc^{fl/fl}* and *Apc^{fl/fl} Rac1b^{fl/fl}* small intestine. BrdU IHC shown for mice of both genotypes (top panels). Red bar outlines size of proliferative zone. Quantification of BrdU incorporation (bottom panel) (error bars represent SD; Mann-Whitney test; n = 5 v 5).

We also analysed tumour numbers across different areas of the intestinal tract and present this in Figure S3B. We found a significant reduction of tumour number in the duodenum and jejunum but not in the ileum or colon. Interestingly, when we analysed *Rac1b* expression in colonic tumours we found it to be present at levels higher than that of normal tissue in the majority of colonic tumours suggesting inefficient deletion of *Rac1b* in the colon may be the reason for this (Figure S3D). These data are outlined on page 5 in the 5th and 6th highlighted sections.

4. Effect of RAC1B on EGFR signaling.

BioID experiments suggest that RAC1B interacts with EGFR signaling components and biochemical studies and antisense studies in cell lines and organoids are consistent with this sort of pathway.

However, no evidence is provided from the animal models to support this. At a minimum, one would like to see staining for phosphor-EGFR and phospho-ERK to demonstrate that deletion of RAC1B modulates EGFR signaling *in vivo*.

To determine whether *Rac1b* deletion modulates EGFR signalling *in vivo* we analysed pEGFR and pERK levels in the tumours derived from these mice (Figure S5H-I). We find that pEGFR levels are significantly lower in the *Rac1b* deficient tumours but pERK levels are unchanged. This suggests that even as tumours establish and develop, EGFR phosphorylation remains dependent on *Rac1b* expression but alternative mechanisms to activate pERK can be utilised. We also investigated the expression of known EGFR pathway target genes (*Etv4* and *Etv5*). Consistent with the pEGFR IHC expression of both is significantly lower in *Rac1b* deficient tumours (Figure S5J). Data are outlined on page 7, 2nd highlighted section.

5. PDO studies.

The investigators show that knockdown of RAC1B reduces clonogenic growth of PDOs from benign polyps but not from invasive tumors, although the latter could be suppressed by the addition of an EGFR inhibitor. However, the concern here is that this appears to be a single set of tumors, with limited studies and endpoints, and it is not clear if this is a RAC1B high or low tumor. While the morpholino approach is interesting, and seems to be the “therapeutic strategy” highlighted in the title, there is a conspicuous absence of *in vivo* preclinical therapeutic models here. Evidence that RAC1B knockdown in combination with EGFR inhibition *in vivo* is needed to support the title of the study. In addition, the title is a bit confusing; was RAC1B knockdown not effective in EGFR inhibitor “sensitive” colorectal cancers?

The reviewer raises important questions regarding the organoid models we utilised to test the efficacy of *Rac1b* inhibition. To address these concerns we obtained cetuximab resistant human colorectal cancer liver metastatic organoids from the Valeri lab (Vlachogiannis et al., 2018, PMID: 29472484). Two organoid lines were tested, C001 and C002, both are *Kras* WT, both express high levels of *Rac1b* compared to the original tumour utilised (Figure S7F) and both are resistant to cetuximab treatment *in vitro*. In addition C002 was derived from a patient who acquired resistance to cetuximab upon treatment. Treatment of both organoid lines with our *RAC1B* inhibitor sensitised them to cetuximab treatment (Figure 7E-F and S7G-H). Thus, using this highly clinically relevant model we demonstrate the potential for *RAC1B* inhibition to enhance the efficacy of a clinically utilised EGFR inhibitor providing additional evidence to support the conclusions of our manuscript. These data are outlined in the highlighted section on page 9. We agree that testing this combination *in vivo* would be an excellent addition but we have not been able to carry out these experiments at this time. We have reflected this in the text, toning-down our conclusions about the clinical relevance of this approach, and emphasising the fact that our approach is unproven *in vivo* (last highlighted section on page 11).

6. Wnt signaling.

Finally, it is somewhat confusing that RAC1B is both downstream and upstream of Wnt signaling and this does not seem to be sufficiently clarified in the manuscript. How is RAC1B activated by Wnt signaling? And how does it then further enhance the Wnt pathway? The previous publication by Myant et al suggested that ROS and NF- κ B were important pathways downstream of RAC1 but these are not mentioned here. Are they also relevant to RAC1B? The model figure (Fig. 7F) seems to

suggest that RAC1B has two independent actions: (i) enhancing Wnt activity and (ii) enhancing EGFR and ERK activation. Given that previous studies have indicated that EGFR activation can enhance Wnt activation; is it possible that EGFR activation is responsible for the enhanced Wnt activation? If so, then the model figure needs to be revised.

The reviewer raises a number of interesting points regarding the control of Rac1b activation and its function. The splicing of Rac1b has been reported to be controlled by the splicing factor Srsf1 which in turn is a target of Myc. As Myc is a Wnt target gene it is possible that this controls Rac1b splicing but this requires further investigation.

We have investigated how Rac1b enhances Wnt signalling, in particular the potential link between EGFR signalling and Wnt signalling in our model systems. Firstly, we investigated the impact of EGFR inhibition on Wnt target gene expression in *Apc* deficient organoids and found that, similar to following *Rac1b* deletion, a subset of targets were downregulated (Figure S6F). In addition, we depleted *Rac1b* in *Apc* deficient organoids carrying an additional activating mutation of *Kras*. In these organoids, where MAPK signalling is constitutively active, downstream of EGFR, the depletion of *Rac1b* had no effect on organoid clonogenic capacity or Wnt target gene expression (Figure S6G-J). Together, these data suggest there is cross talk between Wnt and EGFR signalling pathways and Rac1b mediated control of EGFR signalling activation is required for efficient Wnt signalling activity. As this occurs in *Apc* deficient organoids it suggests this occurs downstream of beta-catenin nuclear localisation. Defining the precise mechanism of this regulation will be an interesting avenue for future investigation. These data are referred to on page 8 of the new manuscript.

We also investigated the possible involvement of NF-KB and ROS but found these were unaltered in *Rac1b* deficient organoids (Figure S6K-L). Interestingly, we have another manuscript just accepted at Nature Communications investigating the function of RacGEF mediated Rac1 activity in intestinal tumourigenesis. In mice deleted for various RacGEF proteins, ROS levels are reduced suggesting that RacGEF activated Rac1 controls ROS production but Rac1b does not.

In light of the reviewers suggestions and our new data linking EGFR activation to Wnt signalling we have amended our model to take this in to account (Figure S8).

Reviewers' Comments:

Reviewer #1:

Remarks to the Author:

This revised version of the manuscript has been improved by the addition of new data and some additional images. However, I still find some deficiencies that could be addressed before publication.

First, I still find that ISH for RAC1b is very faint and difficult to evaluate.

The relationship between Rab1b depletion and proliferation is also questionable since very few cells seem to be positive for Rac1b and differences in proliferation affect the majority of the tumor mass. In fact, it is difficult to say that tumors shown in Rab1b staining are the same that are shown for BrDU incorporation, and in any case differences are minimal. Better sequential images should be selected to convince readers (and the reviewer) about this conclusion.

Related with Figure 6, authors conclude that there is an increase in the amount of EGFR (in red) loaded into the lysosomes (in green). However, what I see is a clear difference in the total level of green staining in the control panels compared with the Rac1B PMO, with no clear differences in the colocalizing white dots as it is indicated in the graphs. Moreover, in the control and see plenty of white dots that are not indicated by arrows as they are in the Rab1b cells.

In the final figure, and in response to the suggestion of adding more translational data, authors have included a couple of in vitro experiment using 2 human organoids. Although images in 7E and 7F are clear, a more quantitative measure of tumor growth such as CellTiter Glo assay or similar would be preferable. I also feel that the inclusion of xenograft in vivo experiments following a similar strategy as in 7E and 7F would help to achieve the required priority for a journal such as Nature Comm

Reviewer #2:

Remarks to the Author:

The authors have addressed all my concerns and the new data are convincing to support the authors' conclusion. There is just one piece of data confusing. In Fig 3H, Apc Rac1b showed defect in organoid formation compared to Apc alone. However, in Fig 7A, in the DMSO control, Apc Rac1b and Apc alone showed no difference in organoids. Can the authors explain this discrepancy?

Reviewer #3:

Remarks to the Author:

No additional comments

We thank the three reviewers again for taking the time to review our revised manuscript and for their insightful analysis and comments. We have outlined our responses below each individual comment in bold. In addition, changes to the manuscript text are referred to in our responses and are highlighted in yellow in the revised manuscript.

Reviewer #1 (Remarks to the Author):

This revised version of the manuscript has been improved by the addition of new data and some additional images. However, I still find some deficiencies that could be addressed before publication.

First, I still find that ISH for Rac1b is very faint and difficult to evaluate.

To aid the interpretation of the RNA Basescope we have included enhanced contrast, false colour images of the images presented in Fig 2D. These can be found in Fig S2D and can be provided for all Basescope experiments if requested. Details of how this was carried out are in the Supplementary Materials and Methods.

The relationship between Rab1b depletion and proliferation is also questionable since very few cells seem to be positive for Rac1b and differences in proliferation affect the majority of the tumor mass.

The reviewer is correct that not all cells show Rac1b Basescope positive foci. However, this is not uncommon for the Basescope technique as it appear to have less sensitivity than standard RNAscope (perhaps due to the short probes used for detecting exon-exon junctions). Indeed, a number of previous studies using Basescope to detect the mutational status of various cancer driver genes also failed to detect signal in every cell within the tumour. For example Kras and Braf mutation in Fig 2 of:

<https://www.nature.com/articles/s41467-017-02295-5>

and PolE mutation in Figure 1C and 1F of:

<https://www.ncbi.nlm.nih.gov/pmc/articles/PMC6738085/>

In fact, it is difficult to say that tumors shown in Rab1b staining are the same that are shown for BrdU incorporation, and in any case differences are minimal. Better sequential images should be selected to convince readers (and the reviewer) about this conclusion.

To aid the interpretation of this data we have included additional overview images to aid this analysis (Fig S3E top panels). As can be seen from these images the tumours analysed for our Basescope and BrdU experiments are the same ones although the reviewer is correct that they are not from sequential images. Unfortunately, we do not have material from these samples left to carry out sequential analysis but have included sequential images from additional tumours stained for Rac1b and BrdU to aid this analysis (Fig S3F).

Related with Figure 6, authors conclude that there is an increase in the amount of EGFR (in red) loaded into the lysosomes (in green). However, what I see is a clear difference in the total level of green staining in the control panels compared with the Rac1B PMO, with no clear differences in the colocalizing white dots as it is indicated in the graphs. Moreover, in the control and see plenty of white dots that are not indicated by arrows as they are in the Rab1b cells.

The reviewer is correct that there are potentially differences in lysosomal localisation / clustering in the *Rac1b* depleted cells. This is an interesting observation, which would be interesting to

investigate in more detail in the future. The reviewer is also correct that there is a level of co-localisation (white dots) in the control panels. The scoring in Fig 6E acknowledges this co-localisation (~7% of EGF-555 foci co-localised with lysosome at 30 mins). We also agree with the reviewer that due to the presence of co-localising foci in the control experiment the inclusion of arrows to indicate co-localised foci does not aid interpretation of the images. We have therefore removed the arrows from the images. However, we respectfully disagree with the reviewer that the level of co-localisation is not different in the *Rac1b* depleted cells. The higher magnification images (Figure S6E) aid the interpretation of this experiment and an increased number of white foci can be observed following *Rac1b* depletion (30 min time point included below). Additionally, our scoring across multiple replicate experiments support the finding that depletion of *Rac1b* increases the co-localisation of EGF-555 and lysotracker (Figure 6E).

In the final figure, and in response to the suggestion of adding more translational data, authors have included a couple of in vitro experiment using 2 human organoids. Although images in 7E and 7F are clear, a more quantitative measure of tumor growth such as CellTiter Glo assay or similar would be preferable. I also feel that the inclusion of xenograft in vivo experiments following a similar strategy as in 7E and 7F would help to achieve the required priority for a journal such as Nature Comm

We have included Resazurin assay measurements from the organoid treatment experiments as requested (Fig S7I-J), confirming our observations that *RAC1B* depletion and cetuximab treatment reduces the viability of these organoid lines. We agree with the reviewer that the inclusion of xenograft experiments would strengthen the potential translational findings of our paper. However, with current working restrictions these experiments would be difficult and time consuming to complete. We feel that our current combination of human cancer analysis, mouse genetic modelling, interactome and mechanistic analysis and direct testing in human tumour organoids provides robust support for our findings. We have also outlined in the discussion the caveat that our targeting strategy requires validation *in vivo*. We would be willing to further temper our conclusions in this regard, and have suggested an alternative the manuscript title to remove reference to potential therapy:

'*RAC1B* mediates WNT and EGFR signalling to promote intestinal tumourigenesis.'

Reviewer #2 (Remarks to the Author):

The authors have addressed all my concerns and the new data are convincing to support the authors' conclusion. There is just one piece of data confusing. In Fig 3H, Apc Rac1b showed defect in organoid formation compared to Apc alone. However, in Fig 7A, in the DMSO control, Apc Rac1b and Apc alone showed no difference in organoids. Can the authors explain this discrepancy?

The reviewer is correct that there is an apparent discrepancy between the findings of these two experiments. This discrepancy is likely explained by the different techniques employed for these experiments. In Figure 7A, we were analysing the effects of EGFR inhibitor on the growth of *fully formed, mature organoids*. Under these conditions, as the organoids are supported by surrounding cells and are adapted to the culture conditions only large effects on organoid growth can be observed. In Figure 3H, digested organoids to single cells and re plated them to carry out a 'clonogenicity' experiment. This assesses the ability of single cells to reform mature organoids and is more representative of the initiation of tumourigenesis. As cells are separated from surrounding, supporting cells this assay is able to detect smaller effects on organoid growth and stem cell activity. Therefore, with this more sensitive assay, we observed a significant impairment on organoid formation following *Rac1b* deletion.

Reviewer #3 (Remarks to the Author):

No additional comments

Reviewers' Comments:

Reviewer #1:

Remarks to the Author:

Authors have appropriately addressed my initial concerns.

Reviewer #2:

Remarks to the Author:

The authors have now addressed all my concerns. No additional comments.